# Task-Agnostic Pre-training and Task-Guided Fine-tuning for Versatile Diffusion Planner

Chenyou Fan [1]  Chenjia Bai [2 3]  Zhao Shan [2 4]  Haoran He [5]  Yang Zhang [2 4]  Zhen Wang [1]

## Abstract

Diffusion models have demonstrated their capabilities in modeling trajectories of multi-tasks. However, existing multi-task planners or policies typically rely on task-specific demonstrations via multi-task imitation, or require task-specific reward labels to facilitate policy optimization via Reinforcement Learning (RL). They are costly due to the substantial human efforts required to collect expert data or design reward functions. To address these challenges, we aim to develop a versatile diffusion planner capable of leveraging large-scale inferior data that contains task-agnostic sub-optimal trajectories, with the ability to fast adapt to specific tasks. In this paper, we propose **SODP**, a two-stage framework that leverages **S**ub-**O**ptimal data to learn a **D**iffusion **P**lanner, which is generalizable for various downstream tasks. Specifically, in the pre-training stage, we train a foundation diffusion planner that extracts general planning capabilities by modeling the versatile distribution of multi-task trajectories, which can be sub-optimal and has wide data coverage. Then for downstream tasks, we adopt RL-based fine-tuning with task-specific rewards to quickly refine the diffusion planner, which aims to generate action sequences with higher task-specific returns. Experimental results from multi-task domains including Meta-World and Adroit demonstrate that SODP outperforms state-of-the-art methods with only a small amount of data for reward-guided fine-tuning.

[1]Northwestern Polytechnical University [2]Institute of Artificial Intelligence (TeleAI), China Telecom [3]Shenzhen Research Institute of Northwestern Polytechnical University [4]Tsinghua University [5]Hong Kong University of Science and Technology. Correspondence to: Chenjia Bai <baicj@chinatelecom.cn>, Zhen Wang <w-zhen@nwpu.edu.cn>.

*Proceedings of the $42^{nd}$ International Conference on Machine Learning*, Vancouver, Canada. PMLR 267, 2025. Copyright 2025 by the author(s).

## 1. Introduction

There has been a long-standing pursuit to develop agents capable of performing multiple tasks (Reed et al., 2022; Lee et al., 2022). Although traditional RL methods have made significant strides in training agents to master individual tasks (Silver et al., 2016; OpenAI et al., 2019), expanding this capability to handle diverse tasks remains a significant challenge due to the diversity of task variants and optimization directions with different rewards. Multi-task RL aims to address this by developing agents via task-conditioned optimization (Yu et al., 2020; Lee et al., 2022) or parameter-compositional learning (Sun et al., 2022; Lee et al., 2023). However, existing approaches often struggle with policy representation since they assume shared latent structures across tasks, which causes issues when tasks have significantly different dynamics and rewards, limiting their ability to capture the multi-modal distribution of optimal behaviors across diverse task spaces. Diffusion models (Sohl-Dickstein et al., 2015; Ho et al., 2020), originally designed for generative tasks, provide a powerful framework to address these difficulties. Their capacity to capture complex, multi-modal distributions within high-dimensional data spaces (Podell et al., 2024; Ho et al., 2022; Jing et al., 2022) makes them well suited to represent the broad variability encountered in multi-task environments.

Motivated by this, existing methods have employed diffusion models to mimic expert behaviors derived from human demonstrations on various tasks (Pearce et al., 2023; Xu et al., 2023; Chi et al., 2023). However, acquiring task-specific demonstrations is often time-consuming and costly, especially in environments requiring specialized domain expertise. Alternative approaches attempt to optimize diffusion models with return guidance (He et al., 2023; Liang et al., 2023) or conventional RL paradigm (Wang et al., 2023), which demands a large volume of data with reward labels for each task. To address the above limitations, we wonder whether a generalized diffusion planner can be learned from a large amount of low-quality trajectories without reward labels, with the ability to adapt quickly to various downstream tasks. We only require the inferior data to comprise a mixture of sub-optimal state-action pairs from various tasks, which can be easily obtained in the real world.

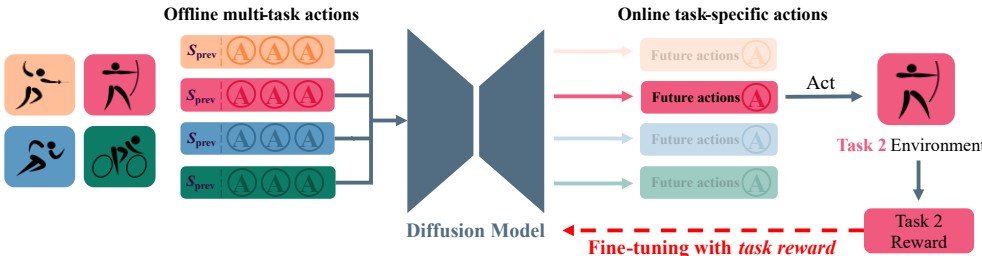

**(a) Stage 1: Pre-training on multiple task datasets**    **(b) Stage 2: Fine-tuning on each specific task**

*Figure 1.* The Overall framework. Different colors represent different tasks. The diffusion model is first pre-trained on a mixed dataset drawn from multiple tasks, and is then fine-tuned for each specific task using task-specific rewards.

In training, the diffusion planner seeks to model the distribution of diverse trajectories with broad coverage, enabling it to acquire generalizable capabilities and allowing the planner to further concentrate on high-reward regions of specific downstream tasks via fast adaptation. An overview of our method is given in Figure 1.

In this paper, we propose a novel framework to utilize **S**ub-**O**ptimal data to train a **D**iffusion **P**lanner (**SODP**) that can generalize across a wide range of downstream tasks. SODP consists of two stages: pre-training and fine-tuning. By leveraging a set of trajectories of different tasks for pre-training, we employ action-sequence prediction to capture shared knowledge across tasks. Since the state space may vary between tasks, focusing on the common action space (e.g., end-effector poses of a robot arm) facilitates task generalization. We frame the pre-training stage as a conditional generative problem that generates future actions based on historical states. Then, inspired by the remarkable success of RL-based alignment for LLMs (Ouyang et al., 2022; Bai et al., 2025), we adopt an RL-based fine-tuning approach to tailor the pre-trained diffusion planner to specific downstream tasks. Specifically, we conduct online interaction based on the pre-trained planner to collect task-specific experiences with reward labels, and perform policy gradients to iteratively refine the predicted action-sequence distribution based on reward feedback of tasks. Through fine-tuning, the diffusion planner can gradually adapt toward generating actions with high task-specific rewards and eventually become optimal for the given task. Figure 2 illustrates our method. In pre-training, the model captures diverse behavior patterns from training data, encompassing inferior and mediocre actions. After fine-tuning, the model shrinks the action distribution and concentrates on generating optimal action sequences for a specific task. Our contributions can be summarized as follows. (i) We propose a novel pre-training and fine-tuning paradigm for learning a versatile diffusion planner, which leverages sub-optimal transitions to capture the broad action distributions across tasks, and adopt task-specific fine-tuning to transfer the planner to downstream tasks. (ii) We give an efficient fine-tuning algo-

rithm based on policy gradient for diffusion planners, which progressively shifts the action distribution to concentrate on regions associated with higher task returns. (iii) We conduct extensive experiments using sub-optimal data from Meta-World (Yu et al., 2019), showcasing its superior performance compared to state-of-the-art approaches.

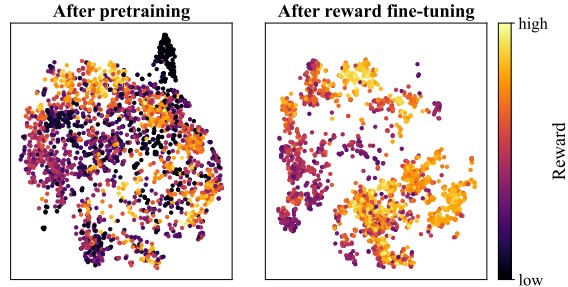

*Figure 2.* Illustration of SODP in Meta-World *button-press-wall* task. We present trajectories generated by the diffusion model after pre-training and fine-tuning of SODP. The pre-trained model captures a wide range of behaviors, and the fine-tuned model discards the inferior behaviors to coverage to high-reward regions.

## 2. Preliminaries

**Multi-task RL** We consider the multi-task RL problem involving $N$ tasks, where each task $\mathcal{T} \sim p(\mathcal{T})$ is represented by a task-specified Markov Decision Process (MDP). Each MDP is defined by a tuple $(\mathcal{S}^{\mathcal{T}}, \mathcal{A}, P^{\mathcal{T}}, R^{\mathcal{T}}, P_0^{\mathcal{T}}, \gamma)$, where $\mathcal{S}^{\mathcal{T}}$ is the state space of task $\mathcal{T}$, $\mathcal{A}$ is the global action space, $P^{\mathcal{T}}(s_{t+1}^{\mathcal{T}}|s_t^{\mathcal{T}}, a_t^{\mathcal{T}}) : \mathcal{S}^{\mathcal{T}} \times \mathcal{A} \to \mathcal{S}^{\mathcal{T}}$ is the transition function, $R^{\mathcal{T}}(s_t^{\mathcal{T}}, a_t^{\mathcal{T}}) : \mathcal{S}^{\mathcal{T}} \times \mathcal{A} \to \mathbb{R}$ is the reward function, $\gamma \in (0, 1]$ is the discount factor, and $P_0^{\mathcal{T}}$ is the initial state distribution. We assume that all tasks share a common action space, executed by the same agent, while differing in their respective reward functions, state spaces, and transition dynamics. At each timestep $t$, the agent perceives a state $s_t^{\mathcal{T}} \in \mathcal{S}^{\mathcal{T}}$, takes an action $a_t^{\mathcal{T}} \in \mathcal{A}$ according to the policy $\pi^{\mathcal{T}}(a_t^{\mathcal{T}}|s_t^{\mathcal{T}})$, and receives a reward $r_t^{\mathcal{T}}$. The agent's objective is to determine an optimal policy that maximizes the expected return across all tasks: $\pi^* = \arg\max_\pi \mathbb{E}_{\mathcal{T} \sim p(\mathcal{T})} \mathbb{E}_{a_t \sim \pi^{\mathcal{T}}} \left[ \sum_{t=0}^{\infty} \gamma^t r_t^{\mathcal{T}} \right]$.

**Diffusion Models** Diffusion models (Sohl-Dickstein et al., 2015) are a type of generative model that first add noise to the data $\boldsymbol{x}_0$ from a unknown distribution $q(\boldsymbol{x}_0)$ in $K$ steps through a forward process defined as:

$$q(\boldsymbol{x}_k|\boldsymbol{x}_{k-1}) := \mathcal{N}(\boldsymbol{x}_k; \sqrt{1-\beta_k}\boldsymbol{x}_{k-1}, \beta_k\boldsymbol{I}), \quad (1)$$

where $\beta_k$ is a predefined variance schedule. Then, a trainable reverse process is constructed as:

$$p_\theta(\boldsymbol{x}_{k-1}|\boldsymbol{x}_k) := \mathcal{N}(\boldsymbol{x}_{k-1}; \mu_\theta(\boldsymbol{x}_k, k), \Sigma_k), \quad (2)$$

where $\mu_\theta(\boldsymbol{x}_k, k)$ is the forward process posterior mean as a function of a noise prediction neural network $\epsilon_\theta(\boldsymbol{x}_k, k)$ with a learnable parameter $\theta$ (Ho et al., 2020). $\epsilon_\theta(\boldsymbol{x}_k, k)$ can be trained via a surrogate loss as

$$\mathcal{L}_{\text{denoise}}(\theta) := \mathbb{E}_{k\sim[1,K], x_0\sim q, \epsilon\sim\mathcal{N}(0,\boldsymbol{I})}\left[\left\|\epsilon - \epsilon_\theta(\boldsymbol{x}_k, k)\right\|^2\right]. \quad (3)$$

After training, samples can be generated by first drawing Gaussian noise $\boldsymbol{x}_K$ and then iteratively denoising $\boldsymbol{x}_K$ into a noise-free output $x_0$ over $K$ iterations using the trained model $\epsilon_\theta(\boldsymbol{x}_k, k)$ by

$$\boldsymbol{x}_{k-1} = \frac{1}{\sqrt{\alpha_k}}\left(\boldsymbol{x}_k - \frac{1-\alpha_k}{\sqrt{1-\bar{\alpha}_k}}\epsilon_\theta(\boldsymbol{x}_k, k)\right) + \sigma_k\mathcal{N}(0, \boldsymbol{I}), \quad (4)$$

where $\alpha_k := 1 - \beta_k$, $\bar{\alpha}_k := \prod_{s=1}^{k}\alpha_s$ and $\sigma_k = \sqrt{\beta_k}$.

## 3. Method

We propose SODP, a two-stage framework that leverages large amounts of sub-optimal data to train a diffusion planner that can generalize to downstream tasks. The process is depicted in Figure 3. In the pre-training stage, we train a guidance-free diffusion model to predict future actions based on historical states, using an mixture offline dataset cross tasks without reward labels. In the fine-tuning stage, we refine the pre-trained model using policy gradient to maximize the task-specific rewards, additionally incorporating a regularization term to prevent the model from losing acquired skills.

### 3.1. Pre-training with Large-scale Sub-optimal Data

Previous works typically model multi-task RL as a conditional generative problem using diffusion models trained on datasets composed of multiple task subsets $\mathcal{D} = \cup_{i=1}^{N}\mathcal{D}_i$, as:

$$\max_\theta \mathbb{E}_{\tau\sim\cup_i\mathcal{D}_i}\left[\log p_\theta(\boldsymbol{x}_0(\tau) \mid \boldsymbol{y}(\tau)\right], \quad (5)$$

which requires additional condition $\boldsymbol{y}(\tau)$ to guide diffusion model to generate desirable trajectories. For instance, $\boldsymbol{y}(\tau)$ should contain the return of trajectory $R(\tau)$ and task description $Z$ as prompt. However, the reward label and trajectory description may be scarce or costly to obtain in the

real-world. To overcome this challenge, we train a diffusion planner that can learn from offline trajectories transitions (i.e., $\{(s_t, a_t, s_{t+1})\}$) without reward label or task descriptions. Specifically, we model the problem as a guidance-free generation process (Chi et al., 2023):

$$\max_\theta \mathbb{E}_{(\boldsymbol{s}_t, \boldsymbol{a}_t)\sim\cup_i\mathcal{D}_i}\left[\log p_\theta(\boldsymbol{a}_t^0 \mid \boldsymbol{s}_t)\right]. \quad (6)$$

Here, we represent $\boldsymbol{x}_0 := \boldsymbol{a}_t^0 = (a_t, a_{t+1}, ..., a_{t+H-1})$ as an action sequence, where $H$ is the planning horizon and $t$ is the timestep sampled from dataset $\mathcal{D}$. We denote $\boldsymbol{s}_t$ as the historical states at timestep $t$ with length $T_o$, i.e., $\boldsymbol{s}_t := \{s_{t-T_o+1}, \ldots s_{t-1}, s_t\}$. The formulation in Eq. (6) enables the model to learn the broad action-sequence distribution of multi-tasks depending on previous observations, without requiring additional guidance. To train our planning model, we modify Eq. (3) to obtain our pre-training objective as:

$$\mathcal{L}_{\text{pre-train}}(\theta) = \mathbb{E}_{k,\epsilon,(\boldsymbol{s}_t, \boldsymbol{a}_t^0)\sim D}\left[\left\|\epsilon - \epsilon_\theta(\boldsymbol{a}_t^k, \boldsymbol{s}_t, k)\right\|^2\right]. \quad (7)$$

Following Eq. (4), we can generate action sequences through a series of denoising steps:

$$\boldsymbol{a}_t^{k-1} = \frac{1}{\sqrt{\alpha_k}}\left(\boldsymbol{a}_t^k - \frac{1-\alpha_k}{\sqrt{1-\bar{\alpha}_k}}\epsilon_\theta(\boldsymbol{a}_t^k, \boldsymbol{s}_t, k)\right) + \sigma_k\mathcal{N}(0, \boldsymbol{I}). \quad (8)$$

Unlike other models, the dataset $\mathcal{D}$ we used for the pre-training stage is not restricted to expert trajectories. As shown in Figure 2, we aim to train a foundation model that captures diverse behaviors and learns general capabilities from inferior trajectories, enabling the planner to enhance its representation and action priors through pre-training before learning on downstream tasks.

### 3.2. Reward Fine-tuning for Downstream Tasks

**MDP notation.** The fine-tuning stage involves two distinct MDPs: one for RL decision process and the other for the diffusion model denoising process. We use the superscript diff (e.g., $s_k^{\text{diff}}$, $a_k^{\text{diff}}$) to denote the MDP associated with diffusion model denoising process, while no superscript is used for the MDP related to the RL process (e.g., $s_t$, $a_t$). Additionally, we use $k \in \{K, \ldots, 0\}$ to represent the diffusion timestep and $t \in \{1, \ldots, T\}$ to represent the trajectory timestep.

We model the denoising process of our pre-trained diffusion planner as a $K$-step MDP as follows:

$$s_k^{\text{diff}} = (s_t, \boldsymbol{a}_t^{K-k}), \qquad a_k^{\text{diff}} = \boldsymbol{a}_t^{K-k-1},$$

$$P_0^{\text{diff}}(s_0^{\text{diff}}) = (\delta_{s_t}, \mathcal{N}(0, I)),$$

$$P^{\text{diff}}(s_{k+1}^{\text{diff}} \mid s_k^{\text{diff}}, a_k^{\text{diff}}) = (\delta_{s_t}, \delta_{a_k^{\text{diff}}}),$$

$$R^{\text{diff}}(s_k^{\text{diff}}, a_k^{\text{diff}}) = \begin{cases} r(s_{k+1}^{\text{diff}}) = r(\boldsymbol{a}_t^0) & \text{if } k = K-1, \\ 0 & \text{otherwise.} \end{cases},$$

$$\pi_\theta^{\text{diff}}(a_k^{\text{diff}} \mid s_k^{\text{diff}}) = p_\theta(\boldsymbol{a}_t^{K-k-1} \mid \boldsymbol{a}_t^{K-k}, \boldsymbol{s}_t), \quad (9)$$

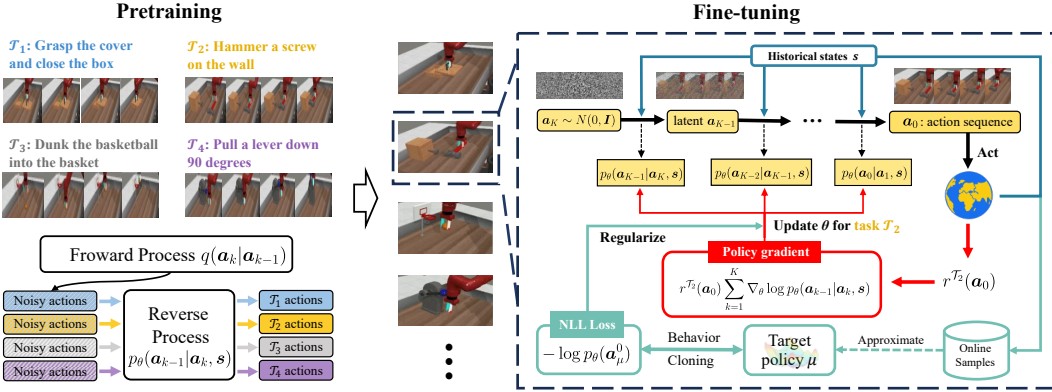

*Figure 3.* Overview of SODP. We initially pre-train a diffusion model using multi-task transition data to predict action sequences from historical states. Subsequently, we fine-tune the model on downstream tasks using policy gradient methods, incorporating a regularization term to mitigate model degradation.

where $s_k^{\text{diff}}$ and $a_k^{\text{diff}}$ are the state and action at timestep $k$, $P_0^{\text{diff}}$ and $P^{\text{diff}}$ are the initial distribution and transition dynamics, $\delta$ is the Dirac delta distribution, $R^{\text{diff}}$ is the reward function and $p_\theta(a_t^{K-k-1} \mid a_t^{K-k}, s_t)$ is the pre-trained diffusion planner. This formulation allows the state transitions in the MDP to be mapped to the denoising process in the diffusion model. The MDP initiates by sampling an initial state $s_0^{\text{diff}} \sim P_0^{\text{diff}}$, which corresponds to sample Gaussian noise $a_t^K$ at the beginning of the reverse process. At each timestep $k$, the policy $\pi_\theta^{\text{diff}}(a_k^{\text{diff}} \mid s_k^{\text{diff}})$ takes an action $a_k^{\text{diff}}$ based on current state $s_k^{\text{diff}}$, which corresponds to generate next latent $a_t^{K-k-1}$ based on current latent $a_t^{K-k}$ following Eq. (8). The reward remains zero until a noise-free output $a_t^0$ is evaluated. Different from previous text-to-image studies that typically evaluate the final sample using a pre-trained reward model (Black et al., 2024; Fan et al., 2024), we aim to fine-tune the pre-trained diffusion planner to maximize rewards of downstream tasks, which makes constructing reward models for all tasks costly. Therefore, we directly evaluate the generated action sequences in an online RL environment for each specific task $\mathcal{T}$. Specifically, for any given timestep $t$, we use the planner to generate future actions $a_t^0 = (a_t, a_{t+1}, \ldots, a_{t+H-1})$ and then execute the first $T_a$ steps. Then we calculate the discounted cumulative reward from the environment to assess the generated sample, expressed as $r(a_t^0) = \sum_t^{T_a} \gamma^{t-1} r^{\mathcal{T}}(s_t, a_t)$. We write $r(a_t^0)$ as shorthand for $r(s_t, a_t^0)$ for brevity.

**Fine-tuning objective.** The objective of fine-tuning our pre-trained diffusion planner is to maximize the expected reward of the generated action sequences for the target downstream task $\mathcal{T}$, which can be defined as:

$$J^{\mathcal{T}}(\theta) = \sum_t \mathbb{E}_{p_\theta(a_t^0 | s_t)}[r^{\mathcal{T}}(a_t^0)]. \tag{10}$$

Directly optimizing the objective $J^{\mathcal{T}}(\theta)$ is intractable since it is infeasible to evaluate the return over all possible actions. Therefore, we utilize policy gradient methods (Sutton

et al., 1999), which estimate the policy gradient and apply a stochastic gradient ascent algorithm for updates. The gradient of the objective $J^{\mathcal{T}}(\theta)$ can be obtained as follows:

$$\nabla_\theta J^{\mathcal{T}}(\theta) =$$
$$\sum_t \mathbb{E}_{p_\theta(a_t^{0:K} | s_t)} \left[ r^{\mathcal{T}}(a_t^0) \sum_{k=1}^K \nabla_\theta \log p_\theta(a_t^{k-1} | a_t^k, s_t) \right]. \tag{11}$$

However, optimizing with Eq. (11) can be computationally intensive, as it requires generating new samples after each optimization step. To enhance sample efficiency and leverage historical sequences, we employ importance sampling, following the approach of proximal policy optimization (PPO) (Schulman et al., 2017), and derive the loss function for reward improvement as follows:

$$\mathcal{L}_{\text{Imp}}^{\mathcal{T}}(\theta) = \sum_t \mathbb{E}_{p_{\theta_{\text{old}}}(a_t^{0:K} | s_t)} \left[ \sum_{k=1}^K -r^{\mathcal{T}}(a_t^0) \cdot \right.$$
$$\left. \max\left( \rho_k(\theta, \theta_{\text{old}}), \text{clip}\left( \rho_k(\theta, \theta_{\text{old}}), 1+\epsilon, 1-\epsilon \right) \right) \right], \quad (12)$$

where $\rho_k(\theta, \theta_{\text{old}}) = \frac{p_\theta(a_t^{k-1} | a_t^k, s_t)}{p_{\theta_{\text{old}}}(a_t^{k-1} | a_t^k, s_t)}$ and $\epsilon$ is a hyperparameter. Then, we can train our model using $\mathcal{L}_{\text{Imp}}^{\mathcal{T}}(\theta)$ in an end-to-end manner, which is equivalent to maximizing the objective in Eq. (10).

**Regularization term.** Fine-tuning the model solely depending on the reward is insufficient since the model may step too far, which can lead to performance collapse and instability during reward maximization. To address this problem, we introduce a Behavior-Clone (BC) regularization term during the fine-tuning process. Concretely, we aim to constrain our policy $\theta$ to closely match a target policy $\mu$, ensuring that $\theta$ does not deviate significantly from $\mu$

after policy updates. This constraint can be modeled using a negative log-likelihood (NLL) loss as:

$$\min_{\theta} \mathbb{E}_{\boldsymbol{a}_{\mu}^0 \sim p_{\mu}} \left[ -\log p_{\theta}(\boldsymbol{a}_{\mu}^0) \right]. \tag{13}$$

Following Ho et al. (2020), we can obtain a surrogate loss to optimize Eq. (13) as follows:

$$\mathcal{L}_{\text{BC}}(\theta) = \mathbb{E}_{k \sim [1,K], \boldsymbol{a}_{\mu}^k \sim p_{\mu}} \left[ \left\| \epsilon(\boldsymbol{a}_{\mu}^k, k) - \epsilon_{\theta}(\boldsymbol{a}_{\mu}^k, k) \right\|^2 \right], \tag{14}$$

where $\epsilon(\boldsymbol{a}_{\mu}^k, k)$ represents the ground-truth noise added to $\boldsymbol{a}_{\mu}^k$ at timestep $k$.

**How to select the target policy?** Intuitively, an ideal target policy is the optimal policy that generates samples $x^*$ satisfying $\mathcal{C}(x^*) \geq \mathcal{C}(x)$ for all possible $x$, where $\mathcal{C}(x)$ represents a measure of the performance or quality of the sample, such as the accumulated reward for action sequences. Since $\mu$ is unknown during fine-tuning, we approximate it by sampling action sequences $\boldsymbol{a}$ that satisfy $\mathcal{C}(\boldsymbol{a}) \approx \mathcal{C}(\boldsymbol{a}^*)$. In practice, we denote $\boldsymbol{a}^*$ as the best actions from recent play experience, such as those that yielded the highest $n$ rewards or successfully completed the given task. We then sample $\boldsymbol{a}^k$ from these proficient actions obtained from online interaction, nearly equivalent to sampling from $\mu$ to regularize the fine-tuning process. We also remark that the BC regularizer is not the only way to incorporate regularization into Eq. (12). For example, a Kullback–Leibler (KL) divergence between fine-tuned and pre-trained models, or a diffusion pre-train loss can be employed to regularize the fine-tuning process, as shown in text-to-image and text-to-speech generation (Fan et al., 2024; Chen et al., 2024). However, we find these regularization may cause the pre-trained planner trap in sub-optimal regions, hindering performance improvement. We will further discuss them in experiments.

Combining Eq. (12) with Eq. (14), the loss function for reward fine-tuning in downstream tasks $\mathcal{T} \sim p(\mathcal{T})$ is expressed as follows:

$$\mathcal{L}_{\text{fine-tuning}}^{\mathcal{T}}(\theta) = \mathcal{L}_{\text{Imp}}^{\mathcal{T}}(\theta) + \lambda \mathcal{L}_{\text{BC}}(\theta), \tag{15}$$

where $\lambda$ is a weight coefficient. The overall process of pre-training and fine-tuning using SODP is summarized in Alg. 1. Since our goal is to generate complete trajectories rather than individual segments, we utilize a trajectory-level buffer (Zheng et al., 2022) for estimating the target policy $\mu$. Further, to ensure the accuracy of the approximation, we generate several proficient trajectories using the pre-trained model at the beginning of each iteration.

## 4. Related Work

**Diffusion Models in RL.** Diffusion models are a leading class of generative models, achieving state-of-the-art performance across a variety of tasks, such as image generation (Ramesh et al., 2021), audio synthesis (Kong et al.,

2021; Huang et al., 2023), and drug design (Schneuing et al., 2024; Guan et al., 2023). Recent studies have applied them in imitation learning to model human demonstrations and predict future actions (Li et al., 2024; Reuss et al., 2023; He et al., 2024). Other approaches have trained conditional diffusion models either as planners (Ajay et al., 2023; Brehmer et al., 2024; Yuan et al., 2024) or policies (Hansen-Estruch et al., 2023; Kang et al., 2024). However, most of these efforts focus on single-task settings. While some recent works aim to extend diffusion models to multi-task scenarios, they often rely on additional conditions, such as prompts (He et al., 2023) or preference labels (Yu et al., 2024). These methods are limited by their dependence on expert data or explicit task knowledge. In contrast, our method learns broad action-sequence distributions from inferior data to enhance action priors, enabling effective generalization across a range of downstream tasks.

**Fine-tuning Diffusion Models.** Despite the impressive success of diffusion models, they often face challenges in aligning with specific downstream objectives, such as image aesthetics (Schuhmann et al., 2022), fairness (Shen et al., 2024), or human preference (Xu et al., 2024), primarily due to their training on unsupervised data. Some methods have been proposed to address this issue by directly fine-tuning models using downstream objectives (Prabhudesai et al., 2023; Clark et al., 2024), but they rely on differentiable reward models, which are impractical in RL since accurately modeling rewards with neural networks is quite costly (Kim et al., 2023). Other methods reformulate the denoising process as an MDP and apply policy gradients for fine-tuning (Black et al., 2024; Fan et al., 2024). However, they heavily depend on strong pre-trained models and have proven ineffective in our case. Our goal is to fine-tune a less powerful model that has been trained on inferior data.

Concurrent with our work, DPPO (Ren et al., 2025) also explores reward fine-tuning for refining diffusion planners. However, their approach focuses on single-task settings and allows access to expert demonstrations. In contrast, we train our model on multi-task data without the need for superior demonstrations. Additionally, we analyze the limitations of current regularization methods for versatile RL diffusion models and propose a new regularizer that improves the performance of sub-optimal pre-trained models.

## 5. Experiments

In this section, we conduct experiments to evaluate our proposed method and address the following questions: (1) How does the performance of SODP compare to other methods that combine offline pre-training with online fine-tuning? (2) How does the performance of SODP compare to other multi-task learning approaches? (3) How does SODP achieve higher rewards during online fine-tuning?

## 5.1. Experimental Setup

We evaluate SODP in both state-based and image-based environments. We conduct experiments on the Meta-World benchmark (Yu et al., 2019) for both state-based and image-based tasks. We also perform image-based experiments on the Adroit benchmark (Rajeswaran et al., 2018). Image-based experimental results are presented in Appendix C.4.

**Meta-World.** The Meta-World benchmark comprises 50 distinct manipulation tasks, each requiring a Sawyer robot to interact with various objects. These tasks are designed to assess the robot's ability to handle different scenarios, such as grasping, pushing, pulling, and manipulating objects of varying shapes, sizes, and complexities. While the state space and reward functions differ across tasks, the action space remains consistent. Following recent studies (He et al., 2023; Hu et al., 2024), we extend all tasks to a random-goal setting, referred to as MT50-rand.

**Datasets.** Following previous work (He et al., 2023), we use a sub-optimal offline dataset containing 1M transitions for each task. The dataset consists of the first 50% of experiences collected from the replay buffer of an SAC (Haarnoja et al., 2018) agent during training. To verify the applicability of our method to tasks of varying difficulty levels, we divide the entire dataset into four subsets based on the task categories presented in Seo et al. (2023).

**Baselines.** We compare our proposed SODP with several RL baselines that integrate offline pre-training with online fine-tuning: (1) **Cal-QL** (Nakamoto et al., 2024). Employs conservative offline initializations to facilitate online fine-tuning. (2) **IBRL** (Hu et al., 2023). Pre-trains a policy using imitation learning and subsequently fine-tunes it as a RL policy. (3) **RLPD** (Ball et al., 2023). Extends SAC by utilizing a sample buffer that incorporates both offline and online data. All baselines, along with SODP, are pre-trained on the same dataset containing 50M transitions and are subsequently fine-tuned on each task with 1M transitions.

Besides, we compare SODP with the following multi-task RL baselines: (1) **MTSAC.** Extended SAC with one-hot task ID as additional input. (2) **MTBC.** Extended BC to multi-task learning through network scaling and a task-ID-conditioned actor. (3) **MTIQL.** Extended IQL (Kostrikov et al., 2022) with multi-head critic networks and a task-ID-conditioned actor for multi-task policy learning. (4) **MT-DQL.** Extended Diffusion-QL (Wang et al., 2023) which is similar to MTIQL. (5) **MTDT.** Extended Decision Transformer (DT) (Chen et al., 2021a) to multitask settings by incorporating task ID encoding and state inputs for task-specific learning. (6) **Prompt-DT** (Xu et al., 2022). An extension of DT, which generates actions by utilizing trajectory prompts and reward-to-go signals. (7) **MTDIFF** (He et al., 2023). A diffusion-based approach that integrates

Table 1. Average success rate across 3 seeds on Meta-World 50 tasks with random goals (MT50-rand), using sub-optimal data. Each task is evaluated for 50 episodes.

| Method | Meta-World 50 Tasks |
|---|---|
| RLPD | $10.16_{\pm 0.11}$ |
| IBRL | $25.29_{\pm 0.22}$ |
| Cal-QL | $35.09_{\pm 0.12}$ |
| MTSAC | $42.67_{\pm 0.12}$ |
| MTBC | $34.53_{\pm 1.25}$ |
| MTIQL | $43.28_{\pm 0.90}$ |
| MTDQL | $17.33_{\pm 0.03}$ |
| MTDT | $42.33_{\pm 1.89}$ |
| Prompt-DT | $48.40_{\pm 0.16}$ |
| MTDIFF-P | $48.67_{\pm 1.32}$ |
| MTDIFF-P-ONEHOT | $48.94_{\pm 0.95}$ |
| HarmoDT-R | $53.80_{\pm 1.07}$ |
| HarmoDT-M | $57.20_{\pm 0.73}$ |
| HarmoDT-F | $57.20_{\pm 0.68}$ |
| **SODP (ours)** | $\mathbf{60.56}_{\pm 0.14}$ |

Transformer architectures with prompt learning to facilitate generative planning in multitask offline environments. We extend it with a visual extractor in image-based Meta-World experiments. (8) **HarmoDT** (Hu et al., 2024). A DT-based approach that leverages parameter sharing to exploit task similarities while mitigating the adverse effects of conflicting gradients simultaneously. All baselines are trained on the 50M offline transitions.

## 5.2. Main Results

We use the average success rate across all tasks as the evaluation metric and report the mean and standard deviation of success rates across three seeds. As shown in Table 1, our method achieves a 60.56% success rate when learning from inferior data, outperforming all baseline methods. Current offline-online RL methods face challenges in effectively learning from complex, low-quality offline data, resulting in insufficient priors to guide the online fine-tuning. Compared to the existing state-of-the-art multi-task approach, our method demonstrates a 5.9% improvement. Notably, when compared to MTDIFF, the current leading method based on diffusion models, our approach shows a 24.4% improvement. MTBC performs the worst, as imitation learning heavily depends on data quality, and directly cloning behaviors from sub-optimal data typically results in inferior performance. In contrast, our method models versatile action distributions from low-quality data and leverages them as priors to guide policy optimization in downstream tasks, leading to improved performance.

To further analyze the learning dynamics, we sample 10 tasks and present the learning curves of SODP alongside three offline-online baselines. As shown in Figure 4, SODP demonstrates a higher success rate after fine-tuning, whereas

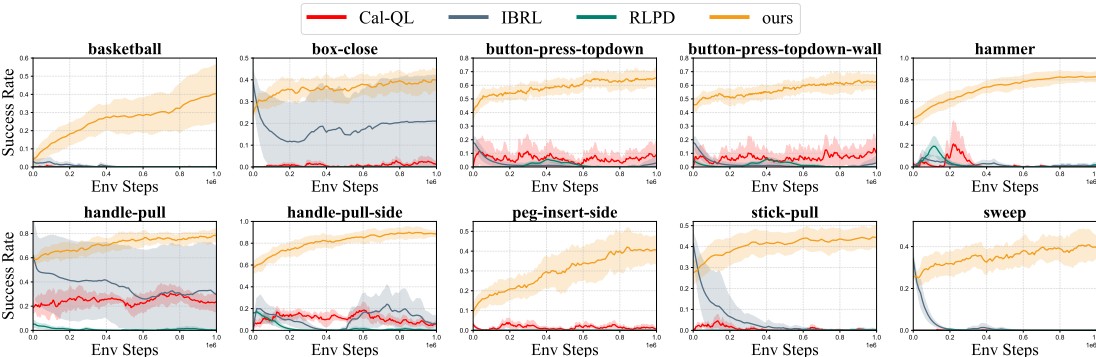

*Figure 4.* Learning dynamics. We sample 10 tasks and present the learning curves of SODP, Cal-QL, IBRL and RLPD across five seeds. X-axis represents environment steps. SODP converges to higher success rates, whereas others struggle with solving the downstream task.

other baseline methods struggle to address the specific task. This performance disparity can be attributed to the inadequate extraction of meaningful guidance during the offline phase. Due to the presence of numerous low-reward trajectories in the sub-optimal dataset, pre-training on such data may misguide the value function. Furthermore, employing a Gaussian policy for behavior cloning fails to capture the multimodal action distributions embedded in the pre-training data. Consequently, this bias can affect the updates during online fine-tuning, potentially trapping the model in a sub-optimal plateau or even degrading its overall performance. The learning curves of SODP, in comparison to other multi-task baselines, are provided in Appendix C.6. We also conduct additional experiments on offline-online baselines, including both diffusion-based and transformer-based methods, with the results presented in Appendix C.3.

### 5.3. Efficiency Validation with Limited Online Samples

To validate the learning efficiency of SODP, we conduct experiments in an extreme scenario where the online fine-tuning steps are reduced to 100k per task. We compare SODP with three offline-online baselines, each using only 100k fine-tuning steps, as well as two leading multi-task RL baselines: MTDIFF and HarmoDT. Since MTDIFF and HarmoDT are purely offline algorithms, we collect online interaction samples during the SODP fine-tuning process. These samples are incorporated as a supplementary dataset alongside the original data, increasing the dataset size from 50M to 50M + 100k×50. We then train both MTDIFF and HarmoDT on this augmented dataset to ensure consistent data usage across our method and the baselines. As shown in Table 2, the performance of our method does not degrade significantly despite a dramatic reduction in fine-tuning steps, demonstrating the strong efficiency of SODP in learning from limited online transitions. In contrast, all three offline-online baselines experience a substantial decline due to insufficient online samples for correcting pre-trained knowl-

edge. For multi-task RL baselines, HarmoDT remains stable when trained on the augmented dataset, whereas the performance of MTDIFF declines markedly. This may be attributed to the increased presence of inferior data collected during the online interaction phase. The generation process of MTDIFF is guided and subsequently biased by these newly introduced low-reward trajectories, whereas the relatively small proportion of low-reward online data compared to the original offline data does not affect the established parameter subspaces in HarmoDT.

*Table 2.* Average success rate of fine-tuning with 100k steps for offline-online methods and training on augmented sub-optimal data for multi-task RL methods.

| Method | Meta-World 50 Tasks |
|---|---|
| RLPD | $7.62_{\pm 0.10}$ |
| IBRL | $17.08_{\pm 0.16}$ |
| Cal-QL | $24.60_{\pm 0.06}$ |
| MTDIFF-P | $27.06_{\pm 0.42}$ |
| HarmoDT-F | $57.37_{\pm 0.34}$ |
| **SODP (ours)** | $\mathbf{59.26}_{\pm 0.18}$ |

### 5.4. Effectiveness of BC Regularization

To demonstrate the effectiveness of our BC regularization, we conduct an ablation study by fine-tuning the same pre-trained model with our BC regularization and other regularization methods. We consider the following variants, each employing a different regularization strategy: (i) **SODP w/o regularization.** This variant fine-tunes the model directly using Eq. (12) without any regularization. (ii) **SODP_kl.** This variant uses a KL regularization term to constrain the divergence between the fine-tuned model and the pre-trained model. (iii) **SODP_pl.** This variant incorporates the original diffusion pre-training loss (PL) into the fine-tuning objective function. The details of these variants are provided in Appendix E, and additional ablation studies on differ-

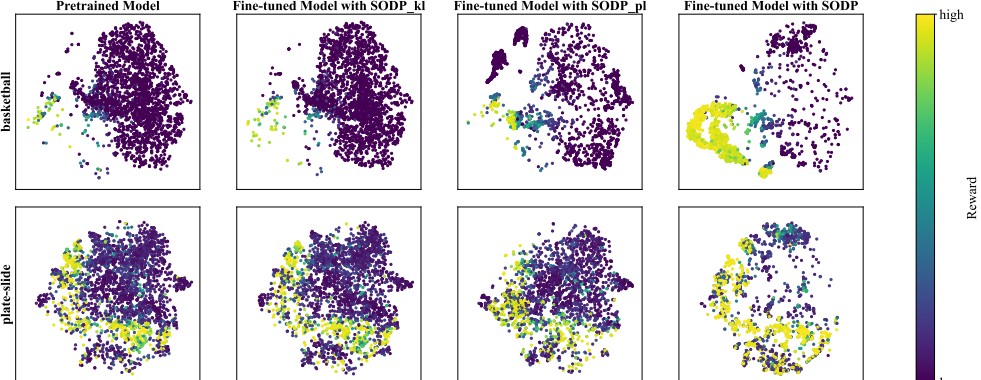

*Figure 5.* Learning dynamics for different regularization. Performance declines without any regularization. Both KL and PL regularization confine the model to sub-optimal regions, while our BC regularization guides the model toward optimal actions.

*Figure 6.* Visualization of trajectories using generated actions for different regularization. KL and PL regularization result in conservative policies, whereas our BC regularization preserves pre-trained knowledge while discovering new actions that lead to high rewards.

ent fine-tuning methods, such as directly fine-tuning with high-quality data, can be found in Appendix C.1.

Figure 5 demonstrates the effectiveness of our regularization in achieving a higher success rate. Directly fine-tuning the model without regularization leads to the worst performance, as the success rate declines due to the model degrading its pre-trained capabilities without constraints. However, adding KL and PL regularization is insufficient, as they cause oscillations near the pre-trained model. We hypothesize that the effectiveness of our BC regularization stems from two factors: (i) it enables the model to reuse acquired skills, preventing performance decline; (ii) it facilitates effective exploration of optimal regions by using optimal $\mu$ as the target policy. To validate this, we visualize trajectories generated by our planner's actions using t-SNE (Van der Maaten & Hinton, 2008). As shown in Figure 6, after fine-tuning with KL and PL regularization, the trajectory distribution remains similar to the pre-training distribution, indicating that the model is reusing learned actions and lacks exploration into new regions. The exploration in PL is unstructured as it may lead to worse regions (e.g., the upper-left region in *basketball*). In contrast, our method demonstrates superior exploration capabilities to discover new, high-reward regions based on acquired knowledge (e.g., the lower-left region in *basketball* and the bottom region in *plate-slide*). Meanwhile, the model can derive

valuable insights from pre-trained knowledge by exploiting discovered high-reward actions (e.g. the central region in *plate-slide*) while discarding low-reward actions.

### 5.5. Effectiveness of Pre-training

We investigate the impact of pre-training by comparing the performance of SODP with a variant that is fine-tuned from scratch using the same loss function defined in Eq. (15), denoted as **SODP_scratch**. Without pre-training, the policy is initially random, which leads to inaccurate estimation of the target policy $\mu$ in the BC regularization term. To mitigate this issue, we initialize the replay buffer with the same initial rollouts used during the fine-tuning of SODP. These rollouts are generated by the pre-trained model, enabling a more accurate approximation of the target policy $\mu$ during early fine-tuning.

We present the learning dynamics of the two methods in Figure 7. Directly fine-tuning the diffusion planner from scratch leads to worse performance. Without pre-training, the planner lacks an action prior to guide its behavior, resulting in stagnation as it struggles to explore high-reward regions. Moreover, the learning process becomes unstable, as the limited useful knowledge is easily overwritten by numerous ineffective trials. We further conduct ablation studies to examine the effect of varying the pre-training data, with results presented in Appendix C.2.

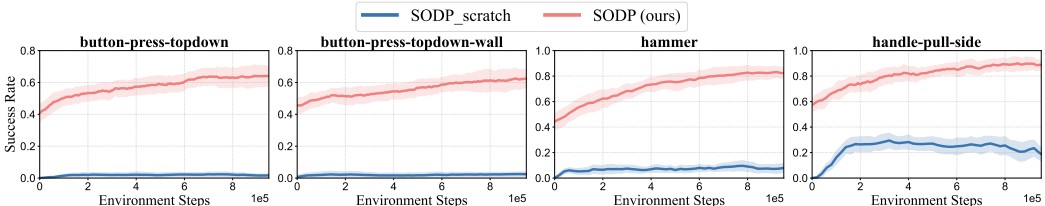

*Figure 7.* Effectiveness of pre-training. X-axis represents environment steps. Fine-tuning from scratch struggles to identify high-reward actions due to the lack of representation prior. In contrast, pre-training allows the planner to extract useful knowledge, guiding fine-tuning by refining the prior distribution towards more effective behaviors.

**Benefits of Multi-Task Pre-Training.** To explore the benefits of pre-training on a multi-task dataset compared to a single-task dataset, we conduct experiments by fine-tuning the model on tasks excluded from the pre-training dataset. Specifically, we pre-train a model on the MT-10 dataset (SODP_mt10) and fine-tune it on three tasks not included in the pre-training data. We compare SODP_mt10 with a variant pre-trained exclusively on the *basketball* dataset (SODP_bas). As shown in Table 3, pre-training on multi-task data improves generalizability to unseen tasks as multi-task data provide a broader range of action distribution priors compared to single-task data.

*Table 3.* Average success rate on unseen tasks.

| Unseen tasks | SODP_mt10 | SODP_bas |
|---|---|---|
| drawer-open | $34.7_{\pm 0.06}$ | $0.0_{\pm 0.0}$ |
| plate-slide-side | $55.3_{\pm 0.33}$ | $0.0_{\pm 0.0}$ |
| handle-pull-side | $71.3_{\pm 0.13}$ | $0.0_{\pm 0.0}$ |

## 6. Conclusion

We propose SODP, a novel framework for training a versatile diffusion planner using sub-optimal data. By effectively combining pre-training and fine-tuning, we capture broad behavioral patterns drawn from large-scale multi-task transitions and then rapidly adapt them to achieve higher performance in specific downstream tasks. During fine-tuning, we introduce a BC regularization method, which preserves the pre-trained model's capabilities while guiding effective exploration. Experiments demonstrate that SODP achieves superior performance across a wide range of challenging manipulation tasks. In future work, we aim to develop embodied versatile agents that can effectively learn to solve real-world tasks using inferior data.

## Acknowledgements

This work is supported by the National Key Research and Development Program of China (Grant No.2024YFE0210900), the National Natural Science Foundation of China (Grant No.62306242), the Young Elite Scientists Sponsorship Program by CAST (Grant No. 2024QNRC001), and the Yangfan Project of the Shanghai (Grant No.23YF11462200).

## Impact Statement

This paper presents work whose goal is to advance the field of Machine Learning. There are many potential societal consequences of our work, none which we feel must be specifically highlighted here.

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

# A. Derivations

## A.1. Derivation of policy gradient in Equation (11)

Assume $p_\theta(\boldsymbol{a}_t^{0:K}|\boldsymbol{s}_t)r^\mathcal{T}(\boldsymbol{a}_t^0)$ and $\nabla_\theta p_\theta(\boldsymbol{a}_t^{0:K}|\boldsymbol{s}_t)r^\mathcal{T}(\boldsymbol{a}_t^0)$ are continuous (Fan et al., 2024), we have:

$$
\begin{aligned}
\nabla_\theta J^\mathcal{T}(\theta) &= \nabla_\theta \sum_t \mathbb{E}_{p_\theta(\boldsymbol{a}_t^0|\boldsymbol{s}_t)}\left[r^\mathcal{T}(\boldsymbol{a}_t^0)\right] \\
&= \sum_t \left[\nabla_\theta \int r^\mathcal{T}(\boldsymbol{a}_t^0) \cdot p_\theta(\boldsymbol{a}_t^0|\boldsymbol{s}_t)d\boldsymbol{a}_t^0\right] \\
&= \sum_t \left[\nabla_\theta \int r^\mathcal{T}(\boldsymbol{a}_t^0) \cdot \left(\int p_\theta(\boldsymbol{a}_t^{0:K}|\boldsymbol{s}_t)d\boldsymbol{a}_t^{1:K}\right) d\boldsymbol{a}_t^0\right] \\
&= \sum_t \left[\int r^\mathcal{T}(\boldsymbol{a}_t^0) \cdot \nabla_\theta \log p_\theta(\boldsymbol{a}_t^{0:K}|\boldsymbol{s}_t) \cdot p_\theta(\boldsymbol{a}_t^{0:K}|\boldsymbol{s}_t) \, d\boldsymbol{a}_t^{0:K}\right] \\
&= \sum_t \left[\int r^\mathcal{T}(\boldsymbol{a}_t^0) \cdot \nabla_\theta \log\left(p_K(\boldsymbol{a}_t^K|\boldsymbol{s}_t)\prod_{k=1}^K p_\theta(\boldsymbol{a}_t^{k-1}|\boldsymbol{a}_t^k,\boldsymbol{s}_t)\right) \cdot p_\theta(\boldsymbol{a}_t^{0:K}|\boldsymbol{s}_t) \, d\boldsymbol{a}_t^{0:K}\right] \\
&= \sum_t \mathbb{E}_{p_\theta(\boldsymbol{a}_t^{0:K}|\boldsymbol{s}_t)}\left[r^\mathcal{T}(\boldsymbol{a}_t^0)\sum_{k=1}^K \nabla_\theta \log p_\theta(\boldsymbol{a}_t^{k-1}|\boldsymbol{a}_t^k,\boldsymbol{s}_t)\right].
\end{aligned}
\tag{16}
$$

## A.2. Derivation of loss function in Equation (12)

By using importance sampling approach, we can rewrite Eq. (16) as follows:

$$
\sum_t \mathbb{E}_{p_{\theta_{\text{old}}}(\boldsymbol{a}_t^{0:K}|\boldsymbol{s}_t)}\left[r^\mathcal{T}(\boldsymbol{a}_t^0)\sum_{k=1}^K \frac{p_\theta(\boldsymbol{a}_t^{k-1}|\boldsymbol{a}_t^k,\boldsymbol{s}_t)}{p_{\theta_{\text{old}}}(\boldsymbol{a}_t^{k-1}|\boldsymbol{a}_t^k,\boldsymbol{s}_t)}\nabla_\theta \log p_\theta(\boldsymbol{a}_t^{k-1}|\boldsymbol{a}_t^k,\boldsymbol{s}_t)\right]
\tag{17}
$$

Then, we can get a new objective function corresponding to Eq. (17) as:

$$
J_{\theta_{\text{old}}}^\mathcal{T}(\theta) = \max_\theta \sum_t \mathbb{E}_{p_{\theta_{\text{old}}}(\boldsymbol{a}_t^{0:K}|\boldsymbol{s}_t)}\left[r^\mathcal{T}(\boldsymbol{a}_t^0)\sum_{k=1}^K \frac{p_\theta(\boldsymbol{a}_t^{k-1}|\boldsymbol{a}_t^k,\boldsymbol{s}_t)}{p_{\theta_{\text{old}}}(\boldsymbol{a}_t^{k-1}|\boldsymbol{a}_t^k,\boldsymbol{s}_t)}\right]
\tag{18}
$$

Let $\rho_k(\theta,\theta_{\text{old}}) = \frac{p_\theta(\boldsymbol{a}_t^{k-1}|\boldsymbol{a}_t^k,\boldsymbol{s}_t)}{p_{\theta_{\text{old}}}(\boldsymbol{a}_t^{k-1}|\boldsymbol{a}_t^k,\boldsymbol{s}_t)}$ denote the probability ratio. Based on PPO (Schulman et al., 2017), we clip $\rho_k$ and use the minimum between the clipped and unclipped ratios to derive a lower bound of the original objective (18), which serves as our final objective function:

$$
J_{\text{clip}}^\mathcal{T}(\theta) = \max_\theta \sum_t \mathbb{E}_{p_{\theta_{\text{old}}}(\boldsymbol{a}_t^{0:K}|\boldsymbol{s}_t)}\left[r^\mathcal{T}(\boldsymbol{a}_t^0)\sum_{k=1}^K \min\Big(\rho_k(\theta,\theta_{\text{old}}),\text{clip}\left(\rho_k(\theta,\theta_{\text{old}}),1+\epsilon,1-\epsilon\right)\Big)\right]
\tag{19}
$$

To refine our pre-trained planner, we employ the negative of objective (19) as the loss function to facilitate reward maximization during fine-tuning.

## A.3. Derivation of loss function in Equation (14)

Directly computing and minimizing the NLL is difficult. However, we can derive an upper bound of Eq. (14) as follows:

$$
\begin{aligned}
\mathbb{E}_{\boldsymbol{a}_\mu^0 \sim p_\mu} \left[ -\log p_\theta(\boldsymbol{a}_\mu^0) \right] &\leq \mathbb{E}_{\boldsymbol{a}_\mu^0 \sim p_\mu} \left[ \mathbb{E}_{q(\boldsymbol{a}_\mu^{1:K}|\boldsymbol{a}_\mu^0)} \left[ -\log \frac{p_\theta(\boldsymbol{a}_\mu^{0:K})}{q(\boldsymbol{a}_\mu^{1:K}|\boldsymbol{a}_\mu^0)} \right] \right] \\
&= \mathbb{E}_{\boldsymbol{a}_\mu^0 \sim p_\mu} \left[ \mathbb{E}_{q(\boldsymbol{a}_\mu^{1:K}|\boldsymbol{a}_\mu^0)} \left[ -\log p(\boldsymbol{a}_\mu^K) - \sum_{k=1}^K \log \frac{p_\theta(\boldsymbol{a}_\mu^{k-1}|\boldsymbol{a}_\mu^k)}{q(\boldsymbol{a}_\mu^k|\boldsymbol{a}_\mu^{k-1})} \right] \right] \\
&= \mathbb{E}_{\boldsymbol{a}_\mu^0 \sim p_\mu} \left[ \sum_{k=2}^K \mathbb{E}_{q(\boldsymbol{a}_\mu^k|\boldsymbol{a}_\mu^0)} D_{\mathrm{KL}}[q(\boldsymbol{a}_\mu^{k-1}|\boldsymbol{a}_\mu^k, \boldsymbol{a}_\mu^0)||p(\boldsymbol{a}_\mu^{k-1}|\boldsymbol{a}_\mu^k)] + \right. \\
&\quad \left. D_{\mathrm{KL}}(q(\boldsymbol{a}_\mu^K|\boldsymbol{a}_\mu^0)||p(\boldsymbol{a}_\mu^K)) - \mathbb{E}_{q(\boldsymbol{a}_\mu^1|\boldsymbol{a}_\mu^0)} \left[ \log p_\theta(\boldsymbol{a}_\mu^0|\boldsymbol{a}_\mu^1) \right] \right]
\end{aligned}
\tag{20}
$$

Following previous work (Ho et al., 2020), the optimization of the bound can be simplified as:

$$
\arg \min_\theta \frac{1}{2\sigma_q^2(k)} \frac{(1-\alpha_k)^2}{(1-\bar{\alpha}_k)\alpha_k} \left\| \epsilon(\boldsymbol{a}_\mu^k, k) - \epsilon_\theta(\boldsymbol{a}_\mu^k, k) \right\|^2
\tag{21}
$$

where:

$$
\sigma_q^2(k) = \frac{(1-\alpha_k)(1-\bar{\alpha}_{k-1})}{1-\bar{\alpha}_k}
\tag{22}
$$

Here, $\epsilon_\theta(\boldsymbol{a}_\mu^k, k)$ is a noise model that learns to predict the source noise $\epsilon(\boldsymbol{a}_\mu^k, k)$ which determines $\boldsymbol{a}_\mu^k$ from $\boldsymbol{a}_\mu^0$.

# B. The Details of SODP

## B.1. Theoretical Motivation

The theoretical motivation of SODP is grounded in the provable efficient reward-free exploration in RL (Wang et al., 2020; Jin et al., 2020). This approach employs a two-stage learning process for developing an RL policy: a reward-free exploration phase and a reward-based optimization phase. The latter stage requires only a limited number of reward labels.

Specifically, in the reward-free stage, the agent optimizes a pure exploration reward, namely, the upper-confidence bound (UCB) term of the value function. Under the linear MDP assumption, we denote the feature of the state-action pair as $\phi(s, a)$, and the value function is a linear mapping of $\phi(s, a)$, expressed as $Q(s, a) = \phi(s, a)^\top \theta$, where $\theta$ is a weight vector. The covariance matrix for state-action pairs is defined as $\Lambda = \sum_i \phi(s_i, a_i)\phi(s_i, a_i)^\top$. Consequently, the UCB-based bonus for reward-free exploration is defined as $b_t = [\phi(s_t, a_t)^\top \Lambda^{-1} \phi(s_t, a_t)]^{1/2}$. Leveraging this bonus, the agent can collect diverse state-action pairs with broad coverage during the exploration stage, thereby providing a sufficient amount of information for the subsequent planning phase. In the second stage, the agent performs least-square value iteration (LSVI), a well-studied RL algorithm, by interacting with the environment. The goal of this stage is to conduct value and policy updates for a specific task with a specific reward function. This two-stage learning paradigm, which combines reward-free exploration and reward-based optimization, can achieve polynomial sample complexity according to theoretical results (Wang et al., 2020). We believe this approach provides a solid theoretical foundation for our method's empirical success.

Our algorithm (i.e., SODP) essentially follows this theoretical motivation but incorporates more practical implementations. Specifically, previous theoretical works have shown that $b_t$ defined in linear MDPs is approximately proportional to the reciprocal pseudo-count of the corresponding state-action pair in the dataset. As a result, the reward-free exploration stage in (Wang et al., 2020) is similar to count-based exploration, which aims to collect diverse datasets in the first stage (Ostrovski et al., 2017). In SODP, we adopt an alternative exploration method (entropy-based exploration) via a non-expert SAC policy to provide sufficient exploration in the data collection stage. We also pre-train a diffusion planner on this exploratory dataset rather than directly performing reward-based policy optimization. This approach seeks to model the distribution of diverse trajectories with broad coverage and provides a well-initialized starting point for the second stage. In Section 5.3, we empirically demonstrate that SODP requires only a very limited number of online samples to achieve strong performance in the second stage. This finding verifies the theoretical results of (Wang et al., 2020; Jin et al., 2020), which suggest that such a two-stage learning paradigm is sample-efficient. We believe that our practical implementation of this theoretical

framework not only enhances the applicability of our method but also maintains the desirable properties of sample efficiency and robustness.

## B.2. Diffusion Policy

We use diffusion policy (Chi et al., 2023) to generate future actions. For any given time step $t$, the model uses the most recent $T_o$ steps of states as input to generate the next $T_p$ action steps. Then, the first $T_a$ steps of these generated actions are executed in the environment without re-planning. In our experiments, we use $T_p = 12, T_o = 2, T_a = 8$ for Meta-World and $T_p = 4, T_o = 2, T_a = 3$ for Adroit.

We employ a CNN-based diffusion policy as our noise model, utilizing a U-net architecture that incorporates Feature-wise Linear Modulation (FiLM) (Perez et al., 2018) to condition on historical states. The implementation is based on the code from `https://github.com/CleanDiffuserTeam/CleanDiffuser`, and we use their default hyper-parameters. For Adroit, we use a simplified backbone provided by Simple DP3 (`https://github.com/YanjieZe/3D-Diffusion-Policy`), which removes some components in the U-net.

## B.3. Implementation Details

The pseudo-code of SODP is given in Alg. 1. We describe details of pre-training and fine-tuning as follows:

- For pretraining, we use cosine schedule for $\beta_k$ (Nichol & Dhariwal, 2021) and set diffusion steps $K = 100$. We pre-train the model for $5e^5$ steps in Meta-Wrold and $3e^3$ steps in Adroit.

- For fine-tuning, we use DDIM (Song et al., 2021) with 10 sampling steps and $\eta = 1$. We fine-tune each task for $1e^6$ steps in Meta-World and $3e^3$ steps in Adroit. Following DPOK (Fan et al., 2024), we perform $p_{\text{step}} \in \{10, 30\}$ gradient steps per episode. We set discount factor $\gamma = 1$ for all tasks.

- We set $N_{\text{init}} \in \{10, 20\}$ for approximating target distribution and $\lambda = 1.0$ as the BC weight coefficient.

- Batch size is set to 256 for both pre-training and fine-tuning.

- We use Adam optimizer (Kingma, 2014) with default parameters for both pre-training and fine-tuning. Learning rate is set to $1e^{-4}$ for pretraining and $1e^{-5}$ for fine-tuning with exponential decay.

---

**Algorithm 1** SODP: Two-stage framework for learning from sub-optimal data

---

**Input:** diffsuion planner $\theta$, $N$ downstream tasks $\mathcal{T}_i$, multi-task sub-optimal data $D = \cup_{i=1}^{N} \mathcal{D}_{\mathcal{T}_i}$, target buffer $\mathcal{B}_{\text{target}}$, replay buffer $\mathcal{B}$, episode length L, pre-train $N_{\text{PT}}$ and fine-tune $N_{\text{FT}}$ steps

`// pre-training model with sub-optimal data`

**for** $t = 1, \ldots, N_{PT}$ **do**

    Sample $(s, a) \sim D$, diffusion time step $k \sim \text{Uniform}(\{1, \ldots, K\})$, noise $\epsilon \sim \mathcal{N}(0, \mathbf{I})$ Update $\theta$ using the loss function (7)

`// fine-tuning model for downstream tasks`

**for** $\mathcal{T}_i \in [\mathcal{T}_1, \ldots, \mathcal{T}_N]$ **do**

    **Initialization:** $\theta \leftarrow \theta_{\text{PT}}$; $\mathcal{B}, \mathcal{B}_{\text{target}} \leftarrow$ Rollout $N_{\text{init}}$ proficient trajectories using $\theta$ **for** $t = 1, \ldots, N_{FT}$ **do**

        **while** *not end of the episode* **do**

            Obtain samples $\boldsymbol{a}_t^{0:K} \sim p_\theta(\boldsymbol{a}_t^{0:K}|s_t)$ Execute the first $T_a$ steps and get reward $r(\boldsymbol{a}_t^0)$ $\mathcal{B} \leftarrow \mathcal{B} \cup (s_t, \boldsymbol{a}_t^{0:K}, r(\boldsymbol{a}_t^0))$

            $s_t \leftarrow s_{t+T_a}, t \leftarrow t + T_a$

        `// approximate target policy` $\mu$

        **if** *proficient* **then**

            $\mathcal{B}_{\text{target}} \leftarrow \mathcal{B}_{\text{target}} \cup \{\boldsymbol{a}_t^{0:K} | t \in \{0, T_a, \ldots, L\}\}$

        Compute $\mathcal{L}_{\text{Imp}}^{\mathcal{T}_i}$ using batches from $\mathcal{B}$ according to Eq. (12) Compute $\mathcal{L}_{\text{BC}}^{\mathcal{T}_i}$ using batches from $\mathcal{B}_{\text{target}}$ according to Eq. (14) Update $\theta$ using the loss function (15)

---

# C. Extended Results

In this section, we provide our full experimental results:

1. Ablation studies evaluating various fine-tuning strategies.

2. Analysis of the impact of pre-training dataset quality.

3. Additional experiments for offline-online baselines

4. Evaluation on image-based environments.

5. Computational efficiency analysis

6. Learning curves of SODP compared to multi-task RL baselines.

## C.1. Ablation studies examining other fine-tuning approaches

To demonstrate the effectiveness of our online fine-tuning approach, we compare it with two alternative fine-tuning methods: (i) **SODP_off**, which involves fine-tuning using high-quality offline data, and (ii) **SODP_off_scratch**, which performs direct training with high-quality data without pre-training. Specifically, we fine-tuned the pre-trained models for 100k steps across five tasks, collecting 200 successful episodes (equivalent to 100k steps) for each task. These datasets were then used to independently train five models in an offline setting, utilizing the same loss function as in Eq. (15) (**SODP_off**). Additionally, to investigate the impact of pre-training on offline fine-tuning, we trained the model directly without pre-training (**SODP_off_scratch**).

The experimental results, presented in Table 4, report the success rates averaged over three seeds. Without pre-training, the model lacks the necessary action priors to efficiently identify high-reward action distributions. Furthermore, directly fine-tuning with high-quality offline data proves insufficient, as static reward labels may fail to provide adequate guidance in dynamic environments, hindering the model's ability to facilitate efficient exploration.

*Table 4.* Average success rate for different fine-tuning approaches.

| Tasks | SODP_off | SODP_off scratch | SODP |
|---|---|---|---|
| button-press-topdown | $58.67_{\pm0.03}$ | $40.67_{\pm0.08}$ | $60.67_{\pm0.03}$ |
| hammer | $71.33_{\pm0.05}$ | $13.33_{\pm0.06}$ | $73.33_{\pm0.03}$ |
| handle-pull-side | $60.67_{\pm0.03}$ | $42.67_{\pm0.08}$ | $81.67_{\pm0.07}$ |
| peg-insert-side | $25.33_{\pm0.03}$ | $0.0_{\pm0.0}$ | $32.67_{\pm0.06}$ |
| handle-pull | $66.67_{\pm0.03}$ | $31.33_{\pm0.06}$ | $75.33_{\pm0.04}$ |
| Average success rate | $56.53_{\pm0.18}$ | $25.6_{\pm0.18}$ | $64.73_{\pm0.19}$ |

To highlight the importance of modeling the diffusion process as a MDP for reward fine-tuning, we consider an alternative approach that directly applies BC during fine-tuning, using only Eq. (14) as the loss function. As shown in Table 5, directly using BC results in poorer performance, as BC lacks reward labels to effectively guide exploration. While BC during the fine-tuning phase enables access to dynamic actions, it is limited to 'imitation' rather than 'evolution,' as the model is unable to differentiate between good and bad actions.

*Table 5.* Average success rate for directly BC during fine-tuning.

| Task | Directly BC | SODP |
|---|---|---|
| button-press-topdown | $51.3_{\pm0.05}$ | $60.7_{\pm0.03}$ |
| basketball | $21.3_{\pm0.03}$ | $41.2_{\pm0.16}$ |
| stick-pull | $26.7_{\pm0.08}$ | $50.5_{\pm0.04}$ |

## C.2. Pre-training using near-optimal data

To evaluate the impact of pre-training data quality on fine-tuning performance, we modified the near-optimal dataset provided by (He et al., 2023) by retaining only the last 50% of the data. This modification ensured that the total number of transitions remained the same as the sub-optimal data used in the main paper, while significantly increasing the proportion of expert trajectories. We refer to this modified dataset as near-optimal data and pre-trained a model on the Meta-World 10 tasks. Subsequently, we followed the same fine-tuning procedure outlined in the main paper to fine-tune the model on each task. The experimental results are presented in Table 6. Incorporating more optimal data during the pre-training stage leads to better performance, as the model gains more priors about the optimal action distributions.

*Table 6.* Average success rate achieved after fine-tuning models pre-trained on different datasets.

| Tasks | Sub-optimal dataset | Near-optimal dataset |
|---|---|---|
| basketball | $52.67_{\pm 0.03}$ | $80.67_{\pm 0.03}$ |
| button-press | $88.00_{\pm 0.02}$ | $89.33_{\pm 0.03}$ |
| dial-turn | $80.67_{\pm 0.02}$ | $74.00_{\pm 0.04}$ |
| drawer-close | $100.00_{\pm 0.00}$ | $100.00_{\pm 0.00}$ |
| peg-insert-side | $62.67_{\pm 0.02}$ | $84.67_{\pm 0.02}$ |
| pick-place | $36.67_{\pm 0.03}$ | $59.33_{\pm 0.03}$ |
| push | $33.33_{\pm 0.03}$ | $50.67_{\pm 0.03}$ |
| reach | $68.67_{\pm 0.05}$ | $95.33_{\pm 0.01}$ |
| sweep-into | $60.67_{\pm 0.03}$ | $75.33_{\pm 0.01}$ |
| window-open | $69.33_{\pm 0.04}$ | $100.0_{\pm 0.00}$ |
| Average success rate | $65.27_{\pm 0.21}$ | $80.93_{\pm 0.16}$ |

## C.3. Additional experiments for offline-online baselines

We consider more baselines that are diffusion-based and support offline pre-training to online RL fine-tuning procedures. Specifically, we extend two diffusion-based methods, DQL (Wang et al., 2023) and IDQL (Hansen-Estruch et al., 2023), to multi-task offline-to-online training on Meta-World 10 tasks, following (Ren et al., 2025). Specifically, we first pretrain a diffusion-based actor using behavior cloning on an offline dataset and then fine-tune both the actor and critic in an online environment. The results are shown in Table 7, the diffusion actor can be misled by inaccurate value network estimations, causing rapid forgetting of actions learned from offline pretraining. Furthermore, since goals in Meta-World are randomized, traditional methods struggle to learn how to complete them effectively.

We also consider transformer-based RL baselines. Specifically, we extend the multi-task Decision Transformer (DT) following (Zheng et al., 2022) by pre-training it on the same sub-optimal data and fine-tuning each task online for 1M steps on the Meta-World 10 tasks. We consider two types of DT: (1) the vanilla DT, which uses a deterministic policy, and (2) the variant proposed in (Zheng et al., 2022), which employs a stochastic policy. The results are presented in Table 7. The stochastic policy improves the success rate by enabling greater exploration in the online environment and discovering more valuable action patterns. However, it still underperforms compared to our method, as the diffusion model can generate more diverse actions.

*Table 7.* Average success rate of MT-10 tasks for diffusion-based and transformer-based offline-online baselines.

| Methods | | Success Rate |
|---|---|---|
| Diffusion-based | MT-DQL | $10.80_{\pm 0.11}$ |
| | MT-IDQL | $13.57_{\pm 0.12}$ |
| Transformer-based | MT-ODT_deter | $40.07_{\pm 0.15}$ |
| | MT-ODT_stoc | $50.47_{\pm 0.21}$ |
| SODP (ours) | | $65.27_{\pm 0.21}$ |

## C.4. Experiments in image-based Meta-World and Adroit

The Adroit benchmark includes three dexterous manipulation tasks, requiring a 24-degree-of-freedom dexterous hand to solve complex challenges such as in-hand manipulation and tool use. The goals in this environment are also randomized. For Adroit, we use images as the observation to assess whether our method can scale to high-dimensional input. To collect offline dataset, we train a VRL3 (Wang et al., 2022) agent for each task and use the initial 30% experiences (90K transitions) from the converged replay buffer.

The action space for different tasks in Adroit is different and is incompatible with MTDIFF and HarmoDT. Therefore, we compare SODP with following baselines designed for complex environments: (1) **BCRNN** (Mandlekar et al., 2021). A variant of BC that employs a Recurrent Neural Network (RNN) as the policy network, predicting the sequence of actions based on the sequence of states as input. (2) **IBC** (Florence et al., 2022). Extended BC with energy-based models (EBM) to train implicit behavioral cloning policies. (3) **Diffusion Policy** (Chi et al., 2023). A diffusion-based approach that predicts future action sequences based on historical states. (4) **DP3** (Ze et al., 2024). A visual imitation learning algorithm that incorporates 3D visual representations into diffusion policies, using a point clouds encoder to process visual observations into visual features. The results for these baselines are directly replicated from those reported in DP3 (Ze et al., 2024).

We scale our method to image-based observations using the Adroit benchmark by employing a point-cloud encoder from DP3 (Ze et al., 2024) to process the 3D scene represented by point clouds. Specifically, we capture depth images directly from the environment and convert them into point clouds using Open3D (Zhou et al., 2018). These point clouds are then processed by the DP3 Encoder, which maps them into visual features. We then train our diffusion planner following the same procedure in Algorithm 1 except the input states are visual features. Following DP3 (Ze et al., 2024), We compute the average of the highest 5 evaluation success rates during training and report the mean and std across 3 seeds. As shown in Table 8, our method achieves an 8.2% improvement across all tasks. Since *hammer* is more challenging than *door*, our method may need more insightful priors from pre-training to achieve better performance.

*Table 8.* Average success rate across 3 seeds on Adroit 3 tasks. IBC and BCRNN are extended by incorporating the DP3 point cloud encoder, resulting in IBD+3D and BCRNN+3D.

| Algorithm \ Task | Adroit Hammer | Door | Pen | **Average** |
|---|---|---|---|---|
| BCRNN | $0_{\pm 0}$ | $0_{\pm 0}$ | $9_{\pm 3}$ | 3.0 |
| BCRNN+3D | $8_{\pm 14}$ | $0_{\pm 0}$ | $8_{\pm 1}$ | 5.3 |
| IBC | $0_{\pm 0}$ | $0_{\pm 0}$ | $9_{\pm 2}$ | 3.0 |
| IBC+3D | $0_{\pm 0}$ | $0_{\pm 0}$ | $10_{\pm 1}$ | 3.3 |
| Diffusion Policy | $48_{\pm 17}$ | $50_{\pm 5}$ | $25_{\pm 4}$ | 31.7 |
| Simple DP3 | $100_{\pm 0}$ | $58_{\pm 4}$ | $46_{\pm 5}$ | 68.0 |
| DP3 | $100_{\pm 0}$ | $62_{\pm 4}$ | $43_{\pm 6}$ | 68.3 |
| SODP (ours) | $67_{\pm 6}$ | $96_{\pm 1}$ | $59_{\pm 4}$ | 73.9 |

We further conduct ablation studies on three Adroit environments, fine-tuning the same pre-trained model with and without BC regularization separately. As shown in Table 9, the performance improvement of the model fine-tuned without BC is marginal, whereas the model fine-tuned with BC achieves significantly better results. This indicates that BC also facilitates learning in high-dimensional single-task settings. This improvement can be attributed to the update stride constraint imposed by BC, which ensures that updates are not too aggressive, thereby preventing the model from becoming trapped in a suboptimal region.

*Table 9.* Ablation study on the image-based Adroit benchmark. We report the average success rate for the pre-trained model, as well as the results of fine-tuning the same model with and without BC regularization.

| Algorithm \ Task | Adroit Hammer | Door | Pen | **Average** |
|---|---|---|---|---|
| Pre-trained model | $20_{\pm 8}$ | $55_{\pm 8}$ | $28_{\pm 3}$ | 34.3 |
| Fine-tuned model w/o BC | $38_{\pm 3}$ | $63_{\pm 3}$ | $30_{\pm 5}$ | 43.7 |
| Fine-tuned model w BC (ours) | $67_{\pm 6}$ | $96_{\pm 1}$ | $59_{\pm 4}$ | 73.9 |

We also conduct experiments on image-based Meta-World 10 tasks. Since no existing image-based sub-optimal dataset for Meta-World is available, we collect data for the 10 tasks by training separate SAC agents for each task, as done in (He et al., 2023), and rendering the environments to obtain image data. We then follow the same procedure as in Adroit to convert the images into point clouds and use the DP3 encoder to extract visual features. For comparison, we consider the following baselines: DP3 and MTDIFF_3D, an extended variant of MTDIFF that employs the same 3D visual encoder used in SODP. The experimental results are presented in Table 10. Our method also outperforms other baselines and achieves a higher success rate, demonstrating its generalizability to complex input modalities.

*Table 10.* Average success rate of image-based MT-10 tasks.

| Methods | Success rate |
|---|---|
| DP3 | $32.6_{\pm0.23}$ |
| MTDIFF_3D | $38.0_{\pm0.82}$ |
| SODP | $47.5_{\pm0.18}$ |

## C.5. Computational efficiency for training and evaluating

We conduct a computational efficiency analysis comparing three offline pre-training followed by online fine-tuning baselines and our method. For training time, we report the average fine-tuning duration per task over 1 million environment steps. For evaluation time, we report the average inference time per task over 50 episodes. All results are obtained using a single NVIDIA RTX 4090 GPU.

*Table 11.* Computational efficiency for olline-online baselines and our method.

| Methods | Fine-tuning time | Inference time |
|---|---|---|
| RLPD | $4.5 - 5, 5h$ | $37s$ |
| IBRL | $6 - 7.5h$ | $39s$ |
| Cal-QL | $8.5 - 10h$ | $40s$ |
| SODP (ours) | $11 - 13h$ | $47s$ |

## C.6. Learning curves of multi-task baselines

We sample 10 tasks and present the learning curves of SODP alongside two leading baselines, MTDIFF and HarmoDT, across five random seeds. The results are shown in Figure 8, where the X-axis represents gradient steps. The first $5e^5$ steps (indicated by the black vertical dashed line) correspond to the pre-training phase of SODP, while the subsequent steps represent the fine-tuning phase (marked by the orange vertical dashed line).

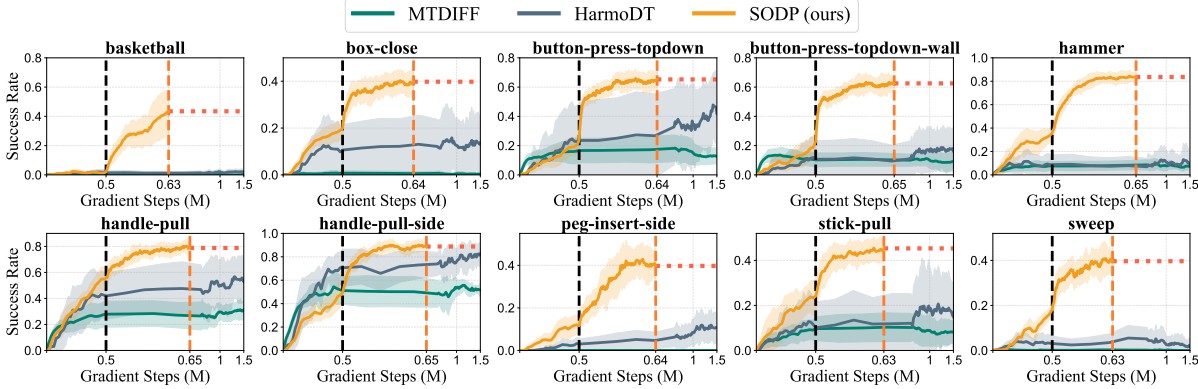

*Figure 8.* Learning dynamics. We sample 10 tasks and present the learning curves of SODP, MTDIFF, and HarmoDT across five seeds. X-axis represents gradient steps. We pre-train the planner on 50M transitions, followed by fine-tuning with 1M transitions per task. SODP rapidly converges to high success rates, whereas MTDIFF and HarmoDT struggle with some challenging tasks.

SODP rapidly converges to high success rates, surpassing the other two baselines. The pre-training stage equips the planner

with comprehensive action distribution priors and allows it to rapidly transfer and enhance these capabilities across a variety of downstream tasks. As a result, the pre-training stage significantly accelerates convergence, leading to more efficient learning in the fine-tuning stage. The two baseline approaches struggle to address complex and challenging tasks such as *basketball* and *hammer*. In contrast, our method effectively guides the model to generate proficient actions, demonstrating the benefits of fine-tuning with policy gradient concerning return maximization. Moreover, while HarmoDT exhibits instability across different random seeds, our method demonstrates robustness against randomness.

## D. The Details of Baselines

We describe the details of baselines used for comparison in our experiments. For offline-online RL, we consider following baselines and use the implementation from https://github.com/irom-princeton/dppo:

- **Cal-QL.** The actor network is implemented as a four-layer MLP and is pre-trained on an offline dataset using the SAC approach. During online fine-tuning, it optimizes conservative value functions that are constrained to exceed the value function of the reference policy.

- **IBRL.** The actor network is implemented as a three-layer MLP and is pre-trained using imitation learning. During fine-tuning, it leverages an ensemble of $N$ critics, randomly selecting two of them for each update step.

- **RLPD.** The actor network is implemented as a three-layer MLP and is pre-trained on an offline dataset using the SAC approach. During online fine-tuning, it leverages an ensemble of $N$ critics, randomly selecting two of them for each update step.

For multi-task RL, we consider following baselines:

- **MTSAC.** The one-hot encoded task ID is incorporated into the original SAC as an additional input.

- **MTBC.** The actor network is modeled using a 3-layer MLP with Mish activation. In training and inference, the scalar task ID is processed through a separate 3-layer MLP with Mish activation to produce a latent variable $z$. The input to the actor network is then formed by concatenating the original state with this latent variable $z$

- **MTIQL.** Similar to MTBC, the actor network incorporates the task ID through a task-aware embedding. A multi-head critic network is employed to estimate the $Q$-values for each task, with each head being parameterized by a 3-layer MLP using Mish activation.

- **MTDQL.** Similar to MTIQL, a multi-head critic network is utilized to predict the Q-value for each task, and the original diffusion actor is extended with an additional task ID input.

- **MTDT.** The task ID is embedded into a latent variable $z$ of size 12. This latent variable is then concatenated with the raw state to form the input tokens.

- **Prompt-DT.** Actions are generated based on trajectory prompts and the reward-to-go. A GPT-2 transformer model is utilized as the noise network.

- **MTDIFF.** Actions are generated by a GPT-based diffusion model that incorporates prompt learning to capture task knowledge. MTDIFF considers a variant: MTDIFF-ONEHOT, which replaces the prompt with a one-hot task ID. We borrow the official codes from https://github.com/tinnerhrhe/MTDiff and use their default hyper-parameters.

- **HarmoDT.** Incorporate trainable task-specific masks to address gradient conflict by identifying an optimal harmony subspace of parameters for each task. There are three variants of HarmoDT: HarmoDT-R, which keeps task masks unchanged; HarmoDT-F and HarmoDT-M utilize different methods to weight masks. We borrow the official codes from https://github.com/charleshsc/HarmoDT and use their default hyper-parameters.

## E. Variants of SODP

In Eq. (15), we introduce a BC regularization term to preserve the pre-trained knowledge and demonstrate its effectiveness compared to existing regularization approaches. We detailed other regularization methods below.

- **SODP w/o regularization.** This variant is similar to DDPO (Black et al., 2024) and DPPO (Ren et al., 2025), which fine-tunes the model directly using Eq. (12) without any regularization.

- **SODP_kl.** This variant is similar to DPOK (Fan et al., 2024), with the addition of a KL regularization term to constrain the divergence between the fine-tuned model and the pre-trained model. Specifically, the regularization term $\mathcal{L}_{\text{KL}}$ used in DPOK is expressed as:

$$\mathcal{L}_{\text{KL}}(\theta) = \sum_{k=1}^{K} \text{KL}(p_\theta(x_{k-1}|x_k)||p_{\text{pre}}(x_{k-1}|x_k)). \tag{23}$$

- **SODP_pl.** This variant is similar to DLPO (Chen et al., 2024), incorporating the original diffusion pre-training loss (PL) into the fine-tuning objective to prevent the model from deviation. Specifically, the regularization term $\mathcal{L}_{\text{PL}}$ used in DLPO is expressed as:

$$\mathcal{L}_{\text{PL}}(\theta) = \mathbb{E}_{k\sim[1,K],p_\theta(x_{1:K})}\left[\|\epsilon(x_k,k) - \epsilon_\theta(x_k,k)\|^2\right]. \tag{24}$$

These methods can be considered as different approaches to selecting the target policy in line with our analysis and can be seen as variants of Eq. (14), where DPOK selects $\mu = \theta_{\text{pre-train}}$ and DLPO selects $\mu = \theta$. The rationale behind their selection is based on the assumption that $\theta \approx \theta_{\text{pre-train}} \approx \theta^*$. This is reasonable for methods like DPOK and DLPO, which utilize pre-trained models such as Stable Diffusion (Rombach et al., 2022) and WaveGrad2 (Chen et al., 2021b). These models already exhibit strong generative capabilities without fine-tuning, and the goal is to make slight adjustments to align them with more fine-grained attributes, such as aesthetic scores and human preferences.

However, this assumption does not apply to our pre-trained planner, as the model is pre-trained on sub-optimal data and lacks the ability to solve complex tasks. We expect it to develop new skills for completing these tasks through fine-tuning. However, directly applying KL regularization to the pre-trained model leads to conservative policies that heavily rely on the existing capability, thereby confining the model to a sub-optimal region. While PL regularization allows some slight exploration, it is uncontrolled. Consequently, as shown in Section 5.4, we observe that the KL regularization almost remains unchanged and the PL regularization slightly increases the performance in *basketball* but decreases in other tasks.

## F. Comparison to DPPO

We summarize some similarities and differences between our work and the concurrent work DPPO (Ren et al., 2025) as follows:

- Both DPPO and our approach formulate the diffusion policy denoising process as an MDP and use policy gradients to fine-tune the model for higher environment rewards.

- DPPO demonstrated that reward-based RL fine-tuning promotes effective exploration, which is consistent with our observations.

- While DPPO requires task-specific expert demonstrations for pre-training, our method pre-trains a foundation model capable of capturing useful behavior patterns from multi-task inferior data.

- We show that directly fine-tuning the pre-trained planner without any regularization, as done in DPPO, fails in the multi-task setting. We further analyze the limitations of current regularization methods and propose a novel BC regularization term. By employing our regularizer, the pre-trained model achieves higher success rates after fine-tuning.

- Unlike DPPO, we don't employ advantage estimator.

# G. Detailed Performance of Offline-Online Methods

We report the success rates of both pre-training and fine-tuning for three offline-online baselines, as well as our method. Specifically, we evaluate each pre-trained model over 50 episodes across three random seeds and present the results in Table 12. The pre-training performance of SODP surpasses that of the other baselines, indicating the advantage of diffusion models in capturing multi-modal action distributions. Furthermore, when fine-tuned with reinforcement learning–based rewards, SODP shows a greater performance improvement compared to traditional offline-online methods, highlighting the effectiveness of our fine-tuning strategy in generalizing more efficiently to downstream tasks.

*Table 12.* Performance of the model after pre-training and fine-tuning for offline-online baselines and our method.

| Methods | After pre-training | After fine-tuning |
|---------|--------------------|--------------------|
| RLPD | $6.40_{\pm 0.03}$ | $10.16_{\pm 0.11}$ |
| IBRL | $14.68_{\pm 0.06}$ | $25.29_{\pm 0.22}$ |
| Cal-QL | $15.82_{\pm 0.06}$ | $35.09_{\pm 0.12}$ |
| SODP (ours) | $31.76_{\pm 0.11}$ | $60.56_{\pm 0.14}$ |

# H. Single-Task Performance

We evaluate the performance for each task for 50 episodes. We report the average evaluated return of pre-trained and fine-tuned models in Table 13.

*Table 13.* Evaluated return of SODP pre-trained model and fine-tuned model for each task in MT50-rand. We report the mean and standard deviation for 50 episodes for each task.

| Tasks | Return of pre-trained model | Return of fine-tuned model |
|:---:|:---:|:---:|
| basketball-v2 | $133.5 \pm 100.7$ | $2347.1 \pm 580.8$ |
| bin-picking-v2 | $96.8 \pm 23.9$ | $602.7 \pm 72.8$ |
| button-press-topdown-v2 | $1405 \pm 20.3$ | $1679 \pm 25.9$ |
| button-press-v2 | $1397 \pm 15.6$ | $2452.7 \pm 89.3$ |
| button-press-wall-v2 | $1375 \pm 10.58$ | $2524.7 \pm 18.1$ |
| coffee-button-v2 | $293.2 \pm 12.1$ | $451.5 \pm 14.4$ |
| coffee-pull-v2 | $39.5 \pm 6.2$ | $117.9 \pm 23.3$ |
| coffee-push-v2 | $33.8 \pm 6.1$ | $273.3 \pm 36.6$ |
| dial-turn-v2 | $1217.7 \pm 239.3$ | $1557.3 \pm 226.7$ |
| disassemble-v2 | $237 \pm 117.2$ | $502 \pm 164.6$ |
| door-close-v2 | $3347.7 \pm 124.9$ | $4116.3 \pm 118.6$ |
| door-lock-v2 | $1042.3 \pm 94.9$ | $2491 \pm 79.5$ |
| door-open-v2 | $2036.3 \pm 79.3$ | $2460.3 \pm 57.7$ |
| door-unlock-v2 | $1335 \pm 46.9$ | $2257.7 \pm 323.0$ |
| hand-insert-v2 | $85.9 \pm 56.7$ | $449.5 \pm 54.9$ |
| drawer-close-v2 | $2468.3 \pm 167.2$ | $3953.7 \pm 214.3$ |
| drawer-open-v2 | $1656 \pm 45.6$ | $2489.7 \pm 188.4$ |
| faucet-open-v2 | $2728.7 \pm 424.1$ | $4094.7 \pm 290.3$ |
| faucet-close-v2 | $2156.7 \pm 113.6$ | $3772 \pm 70.1$ |
| handle-press-side-v2 | $1919.7 \pm 449.5$ | $3478.3 \pm 98.0$ |
| handle-press-v2 | $2216.3 \pm 182.0$ | $3415.7 \pm 221.6$ |
| handle-pull-side-v2 | $1351.7 \pm 119.0$ | $2665.7 \pm 243.9$ |
| handle-pull-v2 | $1510.7 \pm 111.6$ | $2734 \pm 64.3$ |
| lever-pull-v2 | $650.7 \pm 32.5$ | $1068.8 \pm 110.4$ |
| peg-insert-side-v2 | $300.3 \pm 122.2$ | $1969.7 \pm 237.6$ |
| pick-place-wall-v2 | $596.7 \pm 10.6$ | $1175.7 \pm 150.1$ |
| pick-out-of-hole-v2 | $38.5 \pm 6.3$ | $106.7 \pm 7.9$ |
| reach-v2 | $2664.3 \pm 77.5$ | $3083.7 \pm 149.7$ |
| push-back-v2 | $55.8 \pm 26.7$ | $350.9 \pm 30.3$ |
| push-v2 | $46.8 \pm 35.9$ | $148.9 \pm 43.9$ |
| pick-place-v2 | $3.9 \pm 0.2$ | $5.9 \pm 1.8$ |
| plate-slide-v2 | $1268.7 \pm 75.8$ | $2862 \pm 234.5$ |
| plate-slide-side-v2 | $826.4 \pm 54.3$ | $1929.7 \pm 104.5$ |
| plate-slide-back-v2 | $795.8 \pm 62.9$ | $1587.7 \pm 125.0$ |
| plate-slide-back-side-v2 | $626.7 \pm 36.9$ | $1541.3 \pm 78.5$ |
| soccer-v2 | $863.8 \pm 159.5$ | $1234.2 \pm 225.8$ |
| push-wall-v2 | $175.2 \pm 35.0$ | $471.7 \pm 73.9$ |
| shelf-place-v2 | $260.4 \pm 111.5$ | $785.1 \pm 81.5$ |
| sweep-into-v2 | $621.0 \pm 132.9$ | $1282.3 \pm 135.9$ |
| sweep-v2 | $442.3 \pm 83.0$ | $1081.7 \pm 58.0$ |
| window-open-v2 | $1342.3 \pm 60.1$ | $2474.3 \pm 266.4$ |
| window-close-v2 | $1087.7 \pm 78.9$ | $1816.7 \pm 166.3$ |
| assembly-v2 | $282.5 \pm 4.3$ | $446.1 \pm 26.9$ |
| button-press-topdown-wall-v2 | $1374 \pm 16.1$ | $1702.7 \pm 71.5$ |
| hammer-v2 | $1678.3 \pm 52.5$ | $1907.7 \pm 25.4$ |
| peg-unplug-side-v2 | $34.2 \pm 2.9$ | $52.9 \pm 4.6$ |
| reach-wall-v2 | $3373.7 \pm 41.7$ | $3839.7 \pm 69.3$ |
| stick-push-v2 | $412.9 \pm 95.8$ | $833.5 \pm 99.1$ |
| stick-pull-v2 | $1977 \pm 155.9$ | $3116.3 \pm 58.1$ |
| box-close-v2 | $692.3 \pm 22.8$ | $1300.1 \pm 53.2$ |

