# OpenReview forum: "Task-Agnostic Pre-training and Task-Guided Fine-tuning for Versatile Diffusion Planner"
_ICML.cc/2025/Conference — ICML 2025 poster_

### Official Review · Reviewer_4vpc · 2025-03-11

**Overall Recommendation:** 3

**Summary:**

This paper proposes a two-stage framework for training versatile diffusion policies.  The core idea is to learn generalized diffusion policies from a large amount of low-quality multi-task trajectories, followed by task-specific online RL fine-tuning that quickly tailor the pre-trained diffusion policy to the downstream task.  In particular, this paper uses a learning objective, similar to PPO, for fine-tuning the diffusion policy with online interaction.  To enhance the learning stability, this paper further introduces a behavior cloning loss that bias the policy toward the distribution of high-reward actions.  This paper evaluates the proposed method on MetaWorld 50 tasks, outperforming other offline/online RL baselines.

**Claims And Evidence:**

Yes, the claims are supported by the experimental results presented in Table 1 and 2.  Moreover, Table 3 demonstrates the efficacy of multi-task pre-training.

**Essential References Not Discussed:**

I believe the references are complete

**Experimental Designs Or Analyses:**

The experimental designs are reasonable

**Methods And Evaluation Criteria:**

This paper tests baselines and the proposed method with success rate, which is a commonly adopted metric for evaluating robot policies.

**Other Comments Or Suggestions:**

N/A

**Other Strengths And Weaknesses:**

Strengths:

1. One of the core ideas of this paper is to pre-train a generalized diffusion policy with scaled-up sub-optimal training data.  The idea is practically reasonable, since it is much cheaper to collect non-expert trajectories than expert ones.  This paper demonstrates that, by incorporating the knowledge from large-scale sub-optimal trajectories, the pre-trained policy adapts to downstream tasks more effectively and rapidly compared to the policy trained from scratch.

2. Another key insight of this paper to enhance diffusion policies with online RL fine-tuning, which is also technically sound.  It is commonly known that policies learned with behavior cloning have issues with distributional drift, where models perform poorly on unseen scenario, and reinforcement learning could fix these issues.  This paper proposes to enhance diffusion policies, which are often pre-trained with imitation learning, by online RL fine-tuning.  With a simple PPO-like objective and behavior cloning-based regularization, the performance of the diffusion policy is improved by a large margin.

3. This paper is well-written and easy to follow.  I can easily understand the high-level idea and low-level details from the paper.

4. This paper presents good visualization (e.g. Figure 2 and 6), helping readers understand the effect of each component.

---

Weaknesses:

1. This paper lacks the analysis on the effect of separate components.  I believe this paper has three main ideas: 1) pre-training models with large-scale sub-optimal trajectories, 2) fine-tuning models with online RL, and 3) diffusion modeling.  This paper did a good job on analyzing the effect of large-scale pretraining and RL fine-tuning, but does not ablate the effectiveness of diffusion modeling.

More specifically, Table compares to offline RL transformers (HarmoDT), offline RL diffusion models (MTDIFF), and offline+online RL baselines (which often use MLP Gaussian policies).  This paper proposed an offline+online RL diffusion model.  From the comparison to offline RL diffusion models, we know the effectiveness of oneline RL.  From the comparison to offline+online RL baselines, we know diffusion models are better than MLP Gaussian policies.  However, the comparison to offline RL transformers involves two factors (offline RL vs. offline+online RL and transformers vs. diffusion models).  It's unclear for me if the superior performance is mainly resulted from online RL fine-tuning, or the combination of diffusion modeling and online RL.  In other words, would offline+online RL transformers work better?

2. This paper does not present the computational limitation of the proposed method.  From Eqn 12, I suppose gradients are back-propagated across the whole denoising rollout, which may require longer training time and larger memory usage.  It'd be great if the authors could provide analysis on training and testing efficiency, compared to others.

**Questions For Authors:**

1. What is the performance of pre-training a Decision transformer with the same sub-optimal trajectories and fine-tuning it with online RL?

2. What is the training and testing efficiency?

3. I'm wondering if the behavior cloning regularization is needed, if you consider a preference-based RL objectives [1]?

---

Reference:

[1] Direct Preference Optimization: Your Language Model is Secretly a Reward Model. Rafailov et al.

**Relation To Broader Scientific Literature:**

This paper contributes to the field of fine-tuning diffusion generative models, a crucial aspect of image and video generation, as well as chemical and material discovery

**Theoretical Claims:**

The theoretical proof (A.1, A.2 and A.3) in the Appendix looks correct to me.

---

> ### Author Rebuttal · Authors · 2025-04-01
>
> Dear Reviewer 4vpc,
>
> We sincerely appreciate your precious time and constructive comments. In the following, we would like to answer your concerns separately.
>
> **W1**: would offline+online RL transformers work better?
>
> **R1**: We extend the multi-task Decision Transformer (DT) following [1] by pre-training it on the same sub-optimal data and fine-tuning each task online for 1M steps on the Meta-World 10 tasks. We consider two types of DT: (1) the vanilla DT, which uses a deterministic policy, and (2) the variant proposed in [1], which employs a stochastic policy. The results are presented below.
> |Methods|Success rate after pre-training |Success rate after fine-tuning|
> |-|-|-|
> |MT-ODT_deter|24.83 ± 0.10|40.07 ± 0.15|
> |MT-ODT_stoc|37.20 ± 0.12|50.47 ± 0.21|
> |SODP (ours)|31.60 ± 0.14|65.27 ± 0.21|
>
> **W2**: ...gradients are back-propagated across the whole denoising rollout... analysis on training and testing efficiency...
>
> **R2**: Similar to previous work [2], we do not require gradients to be back-propagated through the entire denoising chain at once. From Eq. (11) and (12), each step only requires the final reward $r(a^0)$ to compute the gradient. This allows us to back-propagate the gradient for each step separately, which improves memory efficiency. Furthermore, we employ DDIM sampling during the fine-tuning process, requiring only 10 denoising steps, which reduces both training and inference time. We present the computational efficiency results below. All results are obtained using a single 4090 GPU.
>
> Previous works on fine-tuning text-to-image diffusion models [3] propose a more efficient approach that directly back-propagates the reward gradient through the denoising chain. This method bypasses the need to model the denoising process as a MDP and reduces memory consumption by truncating backpropagation. These reward-backpropagation methods rely on differentiable rewards. However, such differentiable rewards are impractical in RL because reward functions vary across different environments and cannot be easily predicted by a differentiable reward model. Therefore, we use the raw, non-differentiable environment reward and employ PPO for fine-tuning.
>
> Recent studies [4,5] have improved RL-based diffusion model fine-tuning by simplifying the original PPO or introducing dense reward signals at each denoising step, leading to greater effectiveness and better performance. We leave the exploration of optimizing SODP for future work.
>
> |Methods|Fine-tuning time for each task over 1M environment steps| Inference time for each task over 50 episodes|
> |-|-|-|
> |RLPD|4.5 - 5.5 h|~ 37s|
> |IBRL|6 - 7.5 h|~ 39s|
> |Cal-QL|8.5 - 10 h|~ 40s|
> |SODP (ours)|11 - 13 h|~ 47s|
>
> **Q1**: ... the performance of pre-training a Decision transformer with the same sub-optimal trajectories and fine-tuning it with online RL
>
> **R3**: See R1.
>
> **Q2**: What is the training and testing efficiency?
>
> **R4**: See R2.
>
> **Q3**: I'm wondering if the behavior cloning regularization is needed, if you consider a preference-based RL objectives?
>
> **R5**: Preference serves as a weak label when the reward function is hard to obtain and must be converted into an explicit or implicit reward signal [6,7] for model fine-tuning. In our setting, the model interacts with online environments and can readily obtain rewards to evaluate the performance of given actions. Therefore, we can directly fine-tune the model using these ground-truth reward labels without the additional steps of constructing and translating preference labels. The recent success of DeepSeek-R1 [8] further demonstrates the effectiveness of using rewards to improve LLM performance.
> Whether through preferences or rewards, the objective of fine-tuning is a KL-constrained reward maximization problem [6,7], defined as: $L=\mathbb{E} _ {s,a}[R(s,a)]-\beta D _ {\rm KL}[\pi_\theta(a|s)\|\pi_{\rm ref}(a|s)].$
> Preference-based methods such as DPO offer an alternative approach to injecting update gradients into the model but still require constraints to regulate the update step size. However, directly computing the KL-divergence term for diffusion models is not feasible due to their complexity. Therefore, we employ BC-regularization as an alternative to achieve similar effects, ensuring that the learned policy preserves the desirable properties of the reference policy while adapting to the specific task.
>
> [1] Online decision transformer.
>
> [2] DPOK: Reinforcement Learning for Fine-tuning Text-to-Image Diffusion Models.
>
> [3] Directly fine-tuning diffusion models on differentiable rewards.
>
> [4] A Simple and Effective Reinforcement Learning Method for Text-to-Image Diffusion Fine-tuning.
>
> [5] Aligning Few-Step Diffusion Models with Dense Reward Difference Learning.
>
> [6] Training language models to follow instructions with human feedback.
>
> [7] Direct Preference Optimization: Your Language Model is Secretly a Reward Model.
>
> [8] DeepSeek-R1: Incentivizing Reasoning Capability in LLMs via Reinforcement Learning.

---

> > ### Comment · Reviewer_4vpc · 2025-04-06
> >
> > The rebuttal has addressed my concerns, and I'm happy to keep my rating.

---

> > > ### Author Response · Authors · 2025-04-07
> > >
> > > Thank you for your kind response. We sincerely thank you for your time and effort in providing valuable feedback to help us improve our work.

---

### Official Review · Reviewer_aUqS · 2025-03-13

**Overall Recommendation:** 4

**Summary:**

This paper proposes SODP, a two-stage framework for training versatile diffusion planners using sub-optimal data. SODP first pre-trains a foundation model on mixed sub-optimal trajectories without requiring expert demonstrations or reward labels, then employs RL-based fine-tuning with task-specific rewards and behavior cloning regularization. Experiments on Meta-World and Adroit benchmarks demonstrate superior performance over state-of-the-art methods with limited fine-tuning data.

## update after rebuttal
As the rebuttal has addressed most of my concerns. I'd like to keep my score.

**Claims And Evidence:**

The main claims are supported by convincing evidence. SODP achieves 60.56% success rate on Meta-World benchmarks compared to 57.20% for the closest competitor. Ablation studies (especially Appendix C) demonstrate the effectiveness of BC regularization, importance of pre-training, and benefits of the two-stage approach. The efficiency claim is supported by strong performance with just 100k fine-tuning steps.

**Essential References Not Discussed:**

None.

**Experimental Designs Or Analyses:**

The experimental designs are generally sound, with comprehensive comparisons and ablation studies. Visualizations effectively demonstrate how the method improves exploration. Limitations include:
- The computational costs of pre-training (500k steps) are omitted
- No experiments on entirely novel tasks to evaluate true generalization

**Methods And Evaluation Criteria:**

The methods are appropriate, using established benchmarks (Meta-World and Adroit) and standard evaluation metrics (success rate). The approach comprehensively compares against both offline-online RL methods and multi-task RL baselines. The experimental design for testing data efficiency is sensible.

**Other Comments Or Suggestions:**

None.

**Other Strengths And Weaknesses:**

Minor Strengths:
1. Novel use of sub-optimal data
2. Extension to image-based observations.

Minor Weaknesses:
1. Insufficient analysis of scaling with task complexity.

**Questions For Authors:**

1. How does SODP's computational efficiency compare to baselines in terms of training time and inference overhead?
2. From Tab.2, it's interesting to find that with online data collection, the baseline HarmoDT remains stable when trained on the augmented dataset, whereas the performance of MTDIFF declines markedly. How could it happen? Is it because we do not use a good way to distill the new information into the model?
3. Are there specific task categories where SODP struggles that would help identify the advantage of this paper?

**Relation To Broader Scientific Literature:**

The paper builds upon recent works using diffusion models for trajectory planning but innovates by learning from sub-optimal data rather than expert demonstrations. It effectively relates to multi-task RL methods and diffusion model fine-tuning approaches from other domains.

**Theoretical Claims:**

The paper provides theoretical derivations for the policy gradient, loss function, and behavior cloning loss. These derivations appear sound but don't offer novel theoretical insights beyond established techniques from diffusion models and policy optimization literature.

---

> ### Author Rebuttal · Authors · 2025-04-01
>
> Dear Reviewer aUqS,
>
> We sincerely appreciate your precious time and constructive comments. In the following, we would like to answer your concerns separately.
>
> **C1**: The computational costs of pre-training (500k steps) are omitted
>
> **R1**: Thank you for your question. The data used for pre-training are sub-optimal trajectories. Compared to scarce expert demonstrations, collecting sub-optimal data is more cost-effective. Thus, the pre-training stage is data-efficient, allowing us to use relatively inexpensive training data to obtain a reasonable initial policy.
> The pre-training stage is a supervised learning process in which the diffusion model is trained offline. This setup enables the use of data parallelism to efficiently fit the action distribution. In contrast, online interaction is considerably more expensive because data must be collected and used iteratively to optimize the model. Consequently, our primary focus is on the RL fine-tuning process, aiming to improve the efficiency of data utilization and reduce the costs associated with online data collection.
> Overall, large-scale offline data can be leveraged for pre-training, making total computational cost less of a concern. The primary challenge in RL is not the overall computational expense but rather the efficiency of data utilization, as the cost of interaction in the fine-tuning stage is significantly higher.
>
> **C2**: No experiments on entirely novel tasks to evaluate true generalization
>
> **R2**: We fine-tune a model on the *hammer* task and evaluate it on tasks that were not included in the pre-training dataset. The results are presented below.
> ||Pre-trained model | Fine-tuned model on *hammer*|
> |-|-|-|
> |*drawer-close*|54.67 ± 0.06|75.33 ± 0.01|
> |*faucet-open*|19.33 ± 0.01|32.67 ± 0.04|
>
> **C3**: I think Appendix C is very important to support the claims. The authors had better merge them with the results in main paper.
>
> **R3**: Thank you for your suggestion. Appendix C indeed provides further evidence supporting the advantages of our methods. We conduct additional ablation studies on various fine-tuning strategies, such as fine-tuning with high-quality offline data and directly applying behavior cloning on online transitions. Additionally, we fine-tune models pre-trained on different datasets to validate the scalability of our fine-tuning approach. We will merge and discuss these ablation studies in the revised version to enhance clarity.
>
> **W1**: Insufficient analysis of scaling with task complexity.
>
> **R4**: As shown in Figure 8, all methods achieve a high success rate in the simple movement task *handle-pull-side*, which requires the arm to pull a handle upward. However, in more complex two-stage tasks such as *basketball* and *hammer*, which demand precise arm control to interact with small objects, both MTDIFF and HarmoDT exhibit a success rate below 10%.
>
> Despite this challenge, our method consistently outperforms the baselines, achieving higher performance (e.g., 80% success rate in the *hammer* task). This improvement can be attributed to the broad action prior learned during pre-training, which enables effective knowledge transfer across tasks. This is particularly advantageous for challenging tasks where task-specific feedback is limited, requiring the model to leverage skills acquired from other tasks. As shown in Figure 7, fine-tuning from scratch results in a substantial decline in success rate for the *hammer* task, dropping from 80% to 20%.
>
> **Q1**: How does SODP's computational efficiency compare to baselines in terms of training time and inference overhead?
>
> **R5**: Please refer to R2 for Reviewer 4vpc.
>
> **Q2**: the performance of MTDIFF declines markedly. How could it happen? Is it because we do not use a good way to distill the new information into the model?
>
> **R6**: Thank you for your question. We hypothesize that this is due to the 'mode shift' problem identified in previous theoretical work [1,2], which suggests that diffusion models struggle to fit the target distribution when distant modes exist. The original dataset was collected using an SAC agent with a Gaussian policy, while the newly introduced data was collected using our pre-trained model with a diffusion policy. As a result, the data distributions obtained by these policies differ significantly. When combined into a single dataset, the overall distribution contains two distant modes, which may severely hinder the model's ability to accurately capture such complex distributions.
>
> **Q3**: Are there specific task categories where SODP struggles that would help identify the advantage of this paper?
>
> **R7**: As shown in Fig. 8, other baselines often struggle with tasks that require the arm to pick and move a small object (e.g., *basketball*, *sweep*), whereas our method achieves higher performance.
>
> [1] On the Generalization Properties of Diffusion Models. NeurIPS 2023.
>
> [2] Statistical Efficiency of Score Matching: The View from Isoperimetry. arXiv:2210.00726.

---

> > ### Comment · Reviewer_aUqS · 2025-04-07
> >
> > For Q3, my question is that is there any task SODP cannot do well to help identify the contribution of this paper.
> > But all in all, the rebuttal has addressed most of my concerns. I'd like to keep my score.

---

> > > ### Author Response · Authors · 2025-04-07
> > >
> > > Thank you for your response. The main failure cases of SODP occur in challenging tasks that require precise manipulation of extremely small objects, such as the pick-place task, where the arm must pick up and place a puck, and the push task, which requires pushing the puck to a goal. We also observed that other strong baselines, such as HarmoDT, similarly struggle with these tasks.
> > >
> > > Once again, we sincerely thank you for your time and effort in providing valuable feedback to help us improve our work.

---

### Official Review · Reviewer_w6L1 · 2025-03-14

**Overall Recommendation:** 3

**Summary:**

The paper proposes a two-stage framework that pre-trains a diffusion planner using a large number of suboptimal demonstrations and performs task-specific finetuning through reinforcement learning. The diffusion planner is first pre-trained on multi-task offline data of low quality to predict action sequences given historical states. To further optimize performance for particular tasks, the planner is finetuned with policy gradients using task rewards in an online manner. The proposed method achieves strong performance on MetaWorld tasks, outperforming several offline-to-online and multi-task RL baselines.

## Update after rebuttal
I appreciate the authors' response and effort in the additional experiments. Most of my concerns and questions have been addressed. Please include the results above in the final version. I will change my score to 3.

**Claims And Evidence:**

Please see “Other strengths and weaknesses” and “Other Comments Or Suggestions” section below.

**Essential References Not Discussed:**

N/A

**Experimental Designs Or Analyses:**

Please see “Other strengths and weaknesses” and “Other Comments Or Suggestions” section below.

**Methods And Evaluation Criteria:**

Please see “Other strengths and weaknesses” and “Other Comments Or Suggestions” section below.

**Other Comments Or Suggestions:**

1. For offline-to-online settings, it would be helpful to show the performance before and after online finetuning (e.g. in Table 1)

**Other Strengths And Weaknesses:**

Strengths:
- The proposed method effectively leverages a large suboptimal dataset to learn a diverse behavior prior, which can facilitate the second stage of task-specific finetuning
- Extensive ablation studies demonstrate the effectiveness of design choices such as multi-task pretraining and behavior cloning regularization, especially with limited finetuning steps.
- The paper is well-written and easy to read

Weaknesses:
- Different sets of MetaWorld tasks are selected for different ablation studies. It would be helpful to provide some context information or rationale behind the choice of particular tasks, or rigorously keep the same choice of tasks throughout ablation studies.
- Table 1 should include more baselines that (1) are diffusion-based and (2) support offline (pre-)training to online RL fine-tuning procedures for better comparison. The bulk of current baselines are trained offline on suboptimal data without finetuning, and the selected offline-to-online baselines do not use diffusion backbones, which render comparisons unfair.
- It appears that BC regularization is the essential component of the proposed method. It would be helpful to provide ablations on additional environments, such as image-based Adroit, to strengthen the conclusion further

**Questions For Authors:**

1. In MetaWorld, are dense rewards or sparse success signals used for RL finetuning?

**Relation To Broader Scientific Literature:**

N/A

**Theoretical Claims:**

N/A

---

> ### Author Rebuttal · Authors · 2025-04-01
>
> Dear Reviewer w6L1,
>
> We sincerely appreciate your precious time and constructive comments. In the following, we would like to answer your concerns separately.
>
> **W1**: Different sets of MetaWorld tasks are selected for different ablation studies ... provide some context information or rationale behind the choice of particular tasks ...
>
> **R1**: Thank you for your question. Our main experiments were conducted on Meta-World 50 tasks, demonstrating the effectiveness of our method compared to baseline approaches. For ablation studies, such as investigating the impact of different pre-training datasets and evaluating performance on image-based Meta-World, we conducted experiments on Meta-World 10 tasks. This subset is a well-established benchmark that includes tasks of varying difficulty levels and is widely used in multi-task learning research [1,2]. We find it sufficient to assess the effect of the pre-training dataset and the model’s generalization ability to high-dimensional image observations. For other ablation studies, we primarily selected tasks from those used to visualize the learning curves. These tasks were chosen to cover a range of difficulty levels, ensuring a comprehensive comparison of our method's performance.
>
> **W2**: Table 1 should include more baselines that (1) are diffusion-based and (2) support offline re-training to online RL fine-tuning procedures for better comparison
>
> **R2**: We extend two diffusion-based methods, DQL [3] and IDQL [4], to multi-task offline-to-online training on Meta-World 10 tasks, following [5]. Specifically, we first pretrain a diffusion-based actor using behavior cloning on an offline dataset and then fine-tune both the actor and critic in an online environment. However, as the results below show, the diffusion actor can still be misled by inaccurate value network estimations, causing rapid forgetting of actions learned from offline pretraining. Furthermore, since goals in Meta-World are randomized, traditional methods struggle to learn how to complete them effectively.
>
> |Methods|Success rate on Meta-World 10 tasks|
> |-|-|
> |MT-DQL|10.80 ± 0.11|
> |MT-IDQL|13.57 ± 0.13|
> |SODP (ours)|65.27 ± 0.21|
>
> **W3**: ... provide ablations on additional environments, such as image-based Adroit ...
>
> **R3**: Thank you for your suggestion. We conduct ablation studies on three Adroit environments, fine-tuning the same pre-trained model with and without BC regularization separately. As shown below, the performance improvement of the model fine-tuned without BC is marginal, whereas the model fine-tuned with BC achieves significantly better results. This indicates that BC can also facilitate learning in high-dimensional single-task settings. This improvement can be attributed to the update stride constraint imposed by BC, which ensures that updates are not too aggressive, thereby preventing the model from becoming trapped in a suboptimal region [6,7].
>
> ||Pre-trained model |w/o BC| w/ BC|
> |-|-|-|-|
> |Hammer|20 ± 8|38 ± 3|67 ± 6|
> |Door|55 ± 8|63 ± 3|96 ± 1|
> |Pen|28 ± 3|30 ± 5|59 ± 4|
>
> **S1**: For offline-to-online settings ... show the performance before and after online finetuning
>
> **R4**: Thank you for your suggestion. We evaluate the performance of the pre-trained model over 50 episodes across three seeds and report the results below. The pre-training performance of SODP surpasses that of other baselines, demonstrating the advantage of diffusion models in capturing multi-modal action distributions. Additionally, with RL-based reward fine-tuning, the performance improvement exceeds that of traditional offline-online methods, highlighting the effectiveness of our fine-tuning strategy in generalizing to downstream tasks more efficiently.
>
> |Methods|After pre-training| After fine-tuning|
> |-|-|-|
> |RLPD|6.4 ± 0.03|10.16 ± 0.11|
> |IBRL|14.68 ± 0.06|25.29 ± 0.22|
> |Cal-QL|15.82 ± 0.06|35.09 ± 0.12|
> |SODP (ours)|31.76 ± 0.11|60.56 ± 0.14|
>
> **Q1**: In MetaWorld, are dense rewards or sparse success signals used for RL finetuning?
>
> **R5**: We use dense rewards for RL finetuning because it is necessary to provide the reward signal $r(a _ 0)$ for each samling process to guide the update of $p _ \theta$, as shown in Eq.(12). However, as shown in Table 2, we achieve good performance with only 100k online steps, which is 10% of the unlabeled data used for pre-training.
>
> [1] Hard Tasks First: Multi-Task Reinforcement Learning Through Task Scheduling. ICML 2024.
>
> [2] Decomposed Prompt Decision Transformer for Efficient Unseen Task Generalization. NeurIPS 2024.
>
> [3] Idql: Implicit q-learning as an actor-critic method with diffusion policies. arXiv:2304.10573
>
> [4] Diffusion Policies as an Expressive Policy Class for Offline Reinforcement Learning. ICLR 2023.
>
> [5] Diffusion Policy Policy Optimization. arXiv:2409.00588.
>
> [6] Trust Region Policy Optimization. ICML 2015.
>
> [7] Proximal Policy Optimization Algorithms. arXiv:1707.06347.

---

> > ### Comment · Reviewer_w6L1 · 2025-04-09
> >
> > I appreciate the authors' response and effort in the additional experiments. Most of my concerns and questions have been addressed. Please include the results above in the final version. I will change my score to 3.

---

> > > ### Author Response · Authors · 2025-04-09
> > >
> > > Thank you for your kind response. We will incorporate these results in the revised version. Once again, we sincerely thank you for your time and effort in providing valuable feedback to help us improve our work.

---

### Official Review · Reviewer_aKBX · 2025-03-16

**Overall Recommendation:** 4

**Summary:**

The paper presents SODP (Sub-Optimal Diffusion Planner), a two-stage framework that leverages task-agnostic pre-training on sub-optimal trajectory data followed by reward-guided fine-tuning for multi-task reinforcement learning (RL). The approach aims to train a generalizable diffusion-based planner that can extract broad action distributions from large-scale, low-quality offline data and adapt efficiently to new tasks using policy gradient fine-tuning with behavior cloning (BC) regularization. The pre-training phase learns a guidance-free generative model that predicts future actions given past states, without requiring task-specific rewards or expert demonstrations. In the fine-tuning phase, the planner is refined using policy gradients to maximize task-specific returns while maintaining pre-trained knowledge via BC regularization. Empirical evaluations on Meta-World and Adroit show that SODP outperforms state-of-the-art multi-task RL and diffusion-based methods, demonstrating strong sample efficiency, generalization to unseen tasks, and robustness with limited fine-tuning steps. The paper claims that pre-training on diverse sub-optimal data enables the planner to acquire broad action priors, which accelerate task-specific adaptation and achieve higher rewards compared to traditional RL baselines.

**Claims And Evidence:**

The claims in the submission are largely supported by clear empirical evidence, particularly those regarding SODP’s superior performance over state-of-the-art multi-task RL and diffusion-based methods, as demonstrated through success rates in Meta-World, learning curves, and fine-tuning efficiency studies. The claim that BC regularization improves fine-tuning stability is also well-supported by ablation studies comparing different regularization techniques. However, some claims lack sufficient justification. Specifically, the assertion that task-agnostic pre-training on sub-optimal data leads to strong generalization across tasks is not rigorously analyzed beyond empirical results, and the paper does not provide a theoretical explanation or deeper analysis of learned priors. Additionally, while SODP is claimed to be efficient for real-world applications, the paper does not discuss the computational overhead of diffusion-based inference compared to standard policy networks. Lastly, the claim that BC regularization prevents catastrophic forgetting is not explicitly tested through a retention or forgetting analysis. Strengthening these aspects with additional theoretical discussion, computational cost analysis, and empirical studies on task transfer and model forgetting would make the paper’s claims more convincing.

**Essential References Not Discussed:**

The paper includes references and comparison to most of the relevant works in this domain.

**Experimental Designs Or Analyses:**

I reviewed the soundness and validity of the experimental design and analyses presented in the paper. The benchmark selection (Meta-World, Adroit) and success rate evaluation are appropriate for assessing multi-task reinforcement learning performance. The experimental design includes comparisons against strong baselines, such as diffusion-based RL methods (MTDIFF, HarmoDT) and offline-online RL baselines (RLPD, IBRL, Cal-QL), ensuring a fair evaluation. The ablation studies on BC regularization vs. KL and PL regularization, as well as the impact of pre-training on unseen tasks, further strengthen the empirical findings. However, some analyses are missing, such as a computational efficiency comparison and an analysis of forgetting during fine-tuning. While the learning curves and task success rates support the claims, a more detailed breakdown of failure cases and sensitivity analyses would enhance robustness. Overall, the experimental design is strong, but adding efficiency analysis and long-term stability studies would further validate the findings.

**Methods And Evaluation Criteria:**

The proposed methods and evaluation criteria are well-aligned with the problem of multi-task reinforcement learning using diffusion models. The use of Meta-World and Adroit as benchmark environments is appropriate, as they provide diverse manipulation tasks that test generalization and fine-tuning efficiency across different task distributions. The evaluation methodology effectively measures success rates, fine-tuning efficiency, and generalization to unseen tasks, which are key factors in assessing the effectiveness of a multi-task planner. Additionally, the comparison against state-of-the-art RL baselines, including diffusion-based planners (MTDIFF, HarmoDT) and offline-online RL methods (RLPD, IBRL, Cal-QL), ensures a fair assessment of SODP’s performance. However, the evaluation could be further strengthened by including runtime comparisons to assess the computational feasibility of diffusion-based planning versus traditional policy networks. Additionally, while success rate is a relevant metric, analyzing the stability of fine-tuning (e.g., performance degradation over long adaptation periods) and transfer efficiency between tasks would provide deeper insights. Overall, the experimental setup is well-structured, but adding computational efficiency analysis and long-term stability studies would enhance the evaluation.

**Other Comments Or Suggestions:**

N/A

**Other Strengths And Weaknesses:**

Strengths
- The paper demonstrates strong originality in its approach to training a diffusion-based planner using sub-optimal data, rather than relying on expert demonstrations or reward-conditioned learning, which is a notable departure from prior diffusion RL methods
- The proposed two-stage framework (task-agnostic pre-training + reward-guided fine-tuning) is an interesting adaptation of pretraining-finetuning paradigms used in language models and computer vision but applied to multi-task reinforcement learning.
- The integration of behavior cloning (BC) regularization during fine-tuning is also an important contribution, as it effectively prevents policy degradation while allowing adaptation to high-reward behaviors.

Weakness
- While the paper is methodologically strong, there are areas where it could improve in clarity and broader significance.
-  Some explanations, particularly in theoretical motivation for pre-training on sub-optimal data, are not well justified beyond empirical results.
- While the experimental results are compelling, the paper lacks computational efficiency analysis.

**Questions For Authors:**

N/A

**Relation To Broader Scientific Literature:**

The key contributions of the paper build on prior work in diffusion models for reinforcement learning, multi-task RL, and offline-to-online fine-tuning. Diffusion-based models have been used in trajectory generation and planning (e.g. MTDIFF), but these approaches often rely on expert demonstrations or task-conditioned optimization. The paper extends this line of research by proposing a task-agnostic pre-training strategy using sub-optimal data, similar in spirit to unsupervised pre-training in language models, but applied to RL. The fine-tuning approach builds on policy gradient-based RL fine-tuning of generative models, but introduces a behavior cloning (BC) regularization term to preserve pre-trained knowledge while optimizing for task-specific rewards. This aligns with prior findings that pre-trained policies can be refined through RL but may suffer from forgetting or policy collapse without regularization (e.g., RLHF for LLMs, reward-guided fine-tuning of generative models). The work also connects to multi-task RL approaches that aim to learn task-invariant representations (e.g., MTSAC, HarmoDT), but differs by leveraging diffusion models to model broad action distributions without explicit task conditioning. While prior work has explored multi-task pre-training for policy learning, this paper specifically demonstrates that sub-optimal data can serve as a useful training signal for generalization. The findings contribute to the broader literature by showing that diffusion models can learn meaningful priors from noisy data and be efficiently fine-tuned for high-reward adaptation, a paradigm that could be extended to real-world robotic control and hierarchical RL settings.

**Theoretical Claims:**

The paper primarily focuses on an empirical approach rather than providing extensive theoretical claims, but it does include a derivation of policy gradient updates for fine-tuning the diffusion planner in Appendix A. The derivation follows standard policy gradient principles and importance sampling, aligning with established reinforcement learning literature (e.g., PPO and policy optimization methods). The formulation of reward fine-tuning as a K-step Markov Decision Process (MDP) for the diffusion denoising process appears correct in structure, mapping the diffusion process to reinforcement learning updates in a reasonable way. However, the paper does not provide formal theoretical guarantees on why pre-training with sub-optimal data leads to better generalization, which remains an empirical claim rather than a proven result. Additionally, while the policy gradient derivation follows standard RL formulations, it would be useful to include a more detailed discussion of convergence properties or potential biases introduced by the BC regularization term. There were no critical errors in the derivations, but a more formal analysis of pre-training’s impact on generalization and stability would strengthen the theoretical foundation.

---

> ### Author Rebuttal · Authors · 2025-04-01
>
> Dear Reviewer aKBX,
>
> We sincerely appreciate your precious time and constructive comments. In the following, we would like to answer your concerns separately.
>
> **W1**: ...there are areas where it could improve...:
>
> **W1.1**: ... analyzing ... performance degradation over long adaptation periods
>
> **R1.1**: As shown in Tables 1 and 2, increasing the fine-tuning steps from 100k to 1M does not degrade performance. After 1M steps, the learning curve converges and remains stable.
>
> **W1.2**: BC regularization prevents catastrophic forgetting is not explicitly tested ...
>
> **R1.2**: We fine-tune two models on the same *button-press-topdown* task using the same pre-trained model, with and without BC regularization, respectively. We then evaluate them on other tasks from the pre-training dataset. As shown below, the performance of the model fine-tuned with BC regularization does not decline significantly, whereas the model without BC exhibits a noticeable performance drop, indicating that it forgets the acquired skills.
> ||Pre-trained model|w/o BC|w/ BC|
> |-|-|-|-|
> |*button-press-topdown-wall*| 52.67 ± 0.01|21.33 ± 0.05|65.33 ± 0.05
> |*button-press-wall*| 45.33 ± 0.01|20.00 ± 0.07|52.67 ± 0.03
> |*door-close*| 86.00 ± 0.05|47.33 ± 0.06|84.67 ± 0.04
>
> **W1.3**: ...transfer efficiency between tasks
>
> **R1.3**: As shown in R1.2, the model fine-tuned on the *button-press-topdown* task also improves its performance on *button-press-topdown-wall*  and *button-press-wall*, as these tasks share similar goals, allowing the learned skills to be transferred.
>
> **W2**：...theoretical motivation for pre-training on sub-optimal data ...
>
> **R2**：Thank you for your question. The theoretical motivation of our method is grounded in the provable efficient reward-free exploration in RL [1,2]. This approach employs a two-stage learning process for developing an RL policy: a reward-free exploration phase and a reward-based optimization phase.
>
> Specifically, in the reward-free stage, the agent optimizes a purely exploration reward, namely, the upper-confidence bound (UCB) term of the value function. Leveraging a UCB-based bonus, the agent is able to collect diverse state-action pairs with broad coverage, thereby providing sufficient information for the subsequent planning phase. In the second stage, the agent performs least-square value iteration (LSVI) by interacting with the environment. The goal of this stage is to conduct value and policy updates for a specific task with a specific reward function. This two-stage learning paradigm, which combines reward-free exploration and reward-based optimization, can achieve polynomial sample complexity according to theoretical results [1]. We believe this approach provides a solid theoretical foundation for our method's empirical success.
>
> Our algorithm essentially follows this theoretical motivation but incorporates more practical implementations. The reward-free exploration stage in [1] is similar to count-based exploration, which aims to collect diverse datasets in the first stage [3]. In SODP, we adopt an alternative exploration method (entropy-based exploration) via a non-expert SAC policy to provide sufficient exploration in the data collection stage. We also pre-train a diffusion planner on this exploratory dataset rather than directly performing reward-based policy optimization. We empirically demonstrate that SODP requires few online samples to achieve strong performance in the second stage. This finding verifies the theoretical results of [1,2], which suggest that such a two-stage learning paradigm is sample-efficient.
>
> Concerning the BC-regularization, we would like to provide further clarification. We follow the principled RL-tuning formulation used in LLMs fine-tuning [4], which is defined as:
>
> $L=\mathbb{E} _ {s,a}[R(s,a)]-\beta D _ {\rm KL}[\pi_\theta(a|s)\|\pi _ {\rm ref}(a|s)].$
>
> Here, the first term aims to maximize the reward during fine-tuning, while the second term ensures that the learned policy $\pi _ \theta$ remains close to the reference policy $\pi _ {\rm ref}$. In the context of SODP, the $\pi _ {\rm ref}$ is the diffusion planner trained on a large-scale non-expert dataset in the first stage, and $\pi_\theta$ is the policy learned for a specific task in the second stage. However, directly calculating the KL-divergence term for diffusion models is not feasible due to their complex nature. Therefore, we adopt BC-regularization as an alternative to achieve similar effects, ensuring that the learned policy retains the desirable properties of the reference policy while adapting to the specific task.
>
> **W3**：... lacks computational efficiency analysis.
>
> **R3**: Please refer to R2 for Reviewer 4vpc.
>
> [1] On reward-free reinforcement learning with linear function approximation.
>
> [2] Reward-free exploration for reinforcement learning.
>
> [3] Count-based exploration with neural density models.
>
> [4] Training language models to follow instructions with human feedback.

---

> > ### Comment · Reviewer_aKBX · 2025-04-01
> >
> > I am satisfied with the rebuttal and would like to keep my score as is.

---

> > > ### Author Response · Authors · 2025-04-02
> > >
> > > Thank you for your kind response. We sincerely thank you for your time and effort in providing valuable feedback to help us improve our work.

---

### Decision · Program_Chairs · 2025-05-01

**Decision:**

Accept (poster)

**Comment:**

The paper introduces SODP, a novel two-stage framework for learning versatile diffusion planners by leveraging task-agnostic sub-optimal data for pre-training followed by task-guided fine-tuning. Reviewers appreciate the framework for its originality, the practical significance of using sub-optimal data, strong empirical performance on challenging benchmarks, sample efficiency in fine-tuning, and the effectiveness of behavior cloning regularization. However, initial concerns were raised regarding the theoretical grounding for generalization from sub-optimal data, computational efficiency, and the need for more comprehensive comparisons and ablations. Overall, the paper is recommended for acceptance, as the authors' rebuttal effectively addressed most concerns and solidified the work's contribution.